# ZFP36-family RNA-binding proteins in regulatory T cells reinforce immune homeostasis

Beatriz Sáenz-Narciso [1,2], Sarah E. Bell [1,2], Louise S. Matheson [1], Ram K. C. Venigalla[1] & Martin Turner [1] ✉

RNA binding proteins (RBP) of the ZFP36 family limit the differentiation and effector functions of CD4 and CD8 T cells, but little is known of their expression or function in regulatory T (Treg) cells. By using Treg cell-restricted deletion of Zfp36 family members we identify the role of *Zfp36l1* and *Zfp36l2* in Treg cells to maintain immune homeostasis. Mice with Treg cells deficient in these RBP display an inflammatory phenotype with an expansion in the numbers of type-2 conventional dendritic cells, T effector cells, T follicular helper and germinal center B cells and elevated serum cytokines and immunoglobulins. In the absence of *Zfp36l1* and *Zfp36l2*, the pool of cycling CTLA-4 in naïve Treg cells is reduced, Treg cells are less sensitive to IL-2 and IL-7 but are more sensitive to IFNγ. In mice lacking both RBP in Treg cells, the deletion of a single allele of *Ifng* is sufficient to ameliorate the pathology. Our results indicate that ZFP36L1 and ZFP36L2 regulate the availability of IFNγ and are required for the maintenance of Treg cell stability. Thus, ZFP36L1 and ZFP36L2 regulate multiple pathways that enable Treg cells to enforce immune homeostasis.

The Zinc Finger Protein 36 (ZFP36) family of RNA-binding proteins (RBP) are widely expressed and play important roles in developmental biology, stress responses and inflammation[1,2]. They act as repressors of gene expression by direct binding to AU-rich elements in the 3'UTR of mRNAs to trigger RNA degradation[1,2], and, in addition, through mechanisms that remain to be characterized, can repress translation of specific mRNAs or promote translation in subcellular compartments[3–6]. They regulate large numbers of functionally related mRNAs throughout their lifetime creating a system of post-transcriptional operons which regulate immune function[7,8]. The best-characterized operon is the cytokine-encoding mRNAs and their repression by ZFP36 is essential for limiting inflammation. How the other *Zfp36*-family members function and in which cell types they control inflammation and immunity is not well understood.

T lymphocytes transiently express ZFP36 and ZFP36L1 when activated[9,10], and studies using mouse models suggest they each make essential contributions to restraining effector CD4 and CD8 T-cell functions[9–15]. A third family member *Zfp36l2* is expressed by resting T cells and represses cytokine production by memory T cells[4] and by activated naive T cells 48 h after activation[16]. Mice in which all three family members are deleted by *Cd4*-cre at the CD4⁺CD8⁺ thymocyte stage develop hyper-cytokinemia and a lethal inflammatory syndrome[13]. By contrast, mice with *Cd4*-cre-mediated deletion of *Zfp36* and *Zfp36l1* appear healthy[10,13], and show increased resilience following influenza virus infection[10]. Mice with deletion of *Zfp36l1* and *Zfp36l2* in T cells also appear healthy and show reduced pathology and effector T-cell responses following induction of experimental autoimmune encephalomyelitis[13].

As *Cd4*-cre deletes in all TCRαβ⁺ T cells, it remains unclear to what extent these complex phenotypes reflect cell-intrinsic roles of the RBP in effector or Treg cells. Mice which overexpress ZFP36 in all cells have a small increase in the frequency of Treg cells, these were better able to

[1]Immunology Programme, The Babraham Institute, Babraham Research Campus, Cambridge, UK. [2]These authors contributed equally: Beatriz Sáenz-Narciso, Sarah E. Bell. ✉e-mail: martin.turner@babraham.ac.uk

suppress the in vitro proliferation of naive T cells[17]. Another study indicated the potential for ZFP36L2 to be a negative regulator of *Ikzf2* mRNA and to inhibit the suppressive function of induced Treg cells[18] and others have shown that the immunosuppressive cytokine IL-10[19,20] and inhibitory surface receptor CD274/PD-L1[21] are directly repressed by ZFP36-family proteins. Whether the ZFP36-family act in Treg cells to limit or enhance their function is unknown.

Treg cells require the transcription factor FOXP3 for their differentiation and function and have a dominant role in immune tolerance, with roles in tissue homeostasis and resilience to infection[22–24]. They deploy a repertoire of effector functions including the production of soluble factors, contact-mediated depletion of costimulatory molecules and competition with effector T cells for trophic factors and metabolites. Treg cells are particularly sensitive to deprivation of IL-2 which acts via the induction of STAT5 phosphorylation to promote their survival. Treg cells also demonstrate remarkable functional adaptation to local inflammatory environments which can expose them to cytokines, such as IFNγ that can modulate function in a context-dependent manner[25–27]. The extent to which any of these processes are regulated by the ZFP36 family is unknown.

In this work, we employ a conditional deletion strategy in mice to investigate the function of *Zfp36*-family members in Treg cells. The loss of *Zfp36l1* alone in Treg cells results in dysregulation of immune homeostasis, a phenotype that is more severe when combined with deficiency of *Zfp36l2*. We establish that in Treg cells *Zfp36l1* and *Zfp36l2* play a key role in promoting CTLA-4 function to limit the expansion of type-2 conventional dendritic cells (cDC2), in restraining the size of germinal centers (GC), and determine Treg cell sensitivity to IFNγ, IL-2 and IL-7.

## Results

### *Zfp36l1* and *Zfp36l2* are essential in Treg cells to maintain immune homeostasis

To establish if ZFP36-family proteins are expressed in Treg cells we have used mice in which the endogenous allele of each family member has been modified to introduce a fluorescent protein at the site of translation initiation to encode a fusion protein. By identifying Treg cells with surface staining for CD25⁺ and FR4⁺ (Supplementary Fig. S1a), we found mAmetrine–ZFP36 was not detectably expressed by T cells ex vivo. Naive CD44$^{lo}$CD62L$^{hi}$ Treg (nTreg) cells expressed more mCherry–ZFP36L1 and eGFP–ZFP36L2 compared to CD4⁺ CD44$^{lo}$CD62L$^{hi}$ CD25⁻ T (nTconv) cells (Fig. 1a). Furthermore, mCherry–ZFP36L1 was expressed at fourfold greater amounts in CD44$^{hi}$CD62L$^{lo}$ effector Treg (eTreg) compared to nTreg cells, while eGFP–ZFP36L2 was not increased (Fig. 1a). The greater expression of mCherry–ZFP36L1 in eTreg and nTreg cells is consistent with these cells having been recently activated and with ZFP36L1 expression increasing in proportion to TCR signal strength[28] which is known to be greater in naive Treg cells than naive Tconv cells[29].

To establish the requirement for individual ZFP36-family members in Treg cells we deleted *Zfp36*, *Zfp36l1* or *Zfp36l2* using the *Foxp3*$^{YFP-icre}$ (*FYC*) allele which contains the YFP-iCre recombinase fusion protein open reading frame in the 3′UTR of the *Foxp3* gene[30]. Male *Foxp3*$^{YFP-icre}$ *Zfp36*$^{fl/fl}$ (*FYC Zfp36*) and *Foxp3*$^{YFP-icre}$ *Zfp36l2*$^{fl/fl}$ (*FYC l2*) mice were healthy up to twenty weeks of age. By contrast, 9% of *Foxp3*$^{YFP-icre}$ *Zfp36l1*$^{fl/fl}$ male mice (*FYC l1*) had developed clinical signs of marked piloerection, hunched posture with a median age of onset of ten weeks. ZFP36L1 expression in T cells is strongly stimulated by PMA plus ionomycin and we used this to establish whether the protein could be detected in T cells from *FYC l1* mice. ZFP36L1 expression was reduced by tenfold in FOXP3⁺ Treg cells from *FYC l1* mice compared to *FYC* mice but was not differentially expressed in FOXP3⁻ CD4⁺ Tconv cells (Supplementary Fig. 1b), demonstrating efficient and selective deletion of ZFP36L1 in Treg cells. Because naive Treg cells express both ZFP36L1 and ZFP36L2 and *Zfp36l1* is redundant with *Zfp36l2* in diverse cell systems[31–34] we anticipated that the deletion of both genes in Treg

cells would lead to a stronger phenotype than deletion of either gene alone. We thus generated *Foxp3*$^{YFP-icre}$ *Zfp36l1*$^{fl/fl}$*Zfp36l2*$^{fl/fl}$ (referred to as *FYC l1l2*) mice. Deletion of ZFP36L1 in *FYC l1l2* mice was confirmed by flow cytometry to be specific to Treg cells (Supplementary Fig. 1b). Overall, 12% of *FYC l1l2* mice developed clinical symptoms that exceeded the predefined humane endpoint, including marked piloerection, hunched posture, and abdominal distension, with a median age of onset of five weeks. Histological analysis of tissues from *FYC l1l2* mice with clinical symptoms revealed perivascular lymphocytic infiltration into lung and liver and diffuse crypt hyperplasia in the small intestine (Fig. 1b). In the large intestine of *FYC l1l2* mice there was evidence of crypt hyperplasia accompanied by an increase in inflammatory cells in the lamina propria (Fig. 1b). These mice also displayed increased lymph node (LN) cellularity (Fig. 1c), indicating a role for ZFP36L1 and ZFP36L2 in Treg cells to maintain immune homeostasis.

To identify the earliest manifestations of the loss of immune homeostasis, we focused our study on male mice which did not display clinical symptoms. In *FYC l1* males the LN cellularity exceeded that of *FYC* controls and was further elevated in *FYC l1l2* mice (Supplementary Fig. 2a). The proportion and number of YFP⁻ CD44$^{hi}$CD62L$^{lo}$ effector CD4 and CD8 cells within LN (gated as shown in Supplementary Fig. 2b) was increased two- to threefold in *FYC l1* mice, and three- to fivefold in *FYC l1l2* mice compared to *FYC* controls (Fig. 1d). By contrast, in the YFP⁻ CD4⁺ subset from mice lacking *Zfp36* there was no difference in the proportion and only a minor increase in the number of effector cells compared to control mice (1.6-fold) and no change in the CD8⁺ subset (Supplementary Fig. 2c). CD4 and CD8 effector subsets were not different between *FYC l2* and *FYC* mice (Supplementary Fig. 2d). In apparently healthy *FYC l1* and *FYC l1l2* mice Treg cells were biased towards an activated phenotype with a threefold increase in the number of eTreg cells in *FYC l1l2* mice compared to *FYC* control mice (Fig. 1e). In addition, the number of nTreg cells was increased 1.5-fold (Fig. 1f), thus the phenotype of *FYC l1* and *FYC l1l2* mice was not due to a deficit in Treg cell numbers. Furthermore, the combined loss of both *Zfp36l1* and *Zfp36l2* in Treg cells led to a greater expansion of CD4 and CD8 effector cells than loss of *Zfp36l1* alone. Thus, the loss of *Zfp36l1* and *Zfp36l2* in Treg cells leads to a failure of immune homeostasis, which is evident before the appearance of clinical signs.

### Compromised fitness of RBP-deficient Treg cells

In females, *Foxp3* is inactivated on one X chromosome, thus when heterozygous for the *Foxp3*$^{YFP-icre}$ allele, *FYC*$^{/+}$ mice accumulate *icre*-positive and -negative Treg cells which can be distinguished by the expression of YFP. *FYC*$^{/+}$ *l1l2* mice did not develop any clinical symptoms over a period of at least 40 weeks. In *FYC*$^{/+}$ *l1l2* mice, the proportions of eTreg cells amongst YFP⁺ Treg cells were not different than those in *FYC*$^{/+}$ mice; thus, the expansion of eTreg cells in *FYC l1l2* male mice is not a Treg cell-intrinsic phenotype (Supplementary Fig. 3a). Also, we found no difference in the numbers of FOXP3⁻ CD44$^{hi}$CD62L$^{lo}$ CD4 and CD8 cells (Supplementary Fig. 3b). Therefore, *Zfp36l1/Zfp36l2*-deficient Treg cells are insufficient to cause disease when wild-type Treg cells are present. In female *FYC*$^{/+}$ mice the number of FOXP3⁺YFP⁻ Treg cells was twice that of FOXP3⁺YFP⁺ Treg cells in the same mouse suggesting the *Foxp3*$^{YFP-icre}$ allele incurs a minor competitive disadvantage on Treg cells. However, in *FYC*$^{/+}$ *l1l2* mice the *icre*-positive Treg cells were seven times less abundant than *icre*-negative Treg cells in the same mouse (Fig. 2a). Normal Treg cell numbers were detected in the thymus (Supplementary Fig. S3c), thus the competitive disadvantage of *Zfp36l1/Zfp36l2*-deficient Treg cells is revealed in peripheral lymphoid tissue.

To gain insight into the role of *Zfp36l1* and *Zfp36l2* in Treg cell function in the absence of pathology we sorted YFP⁺ CD62L$^{hi}$CD4⁺CD25⁺ cells from female *FYC*$^{/+}$ and *FYC*$^{/+}$ *l1l2* mice (Supplementary Fig. 3d) and performed RNA-seq. Quantitation of reads mapping to the targeted region of *Zfp36l1* and *Zfp36l2* confirmed efficient *icre*-mediated

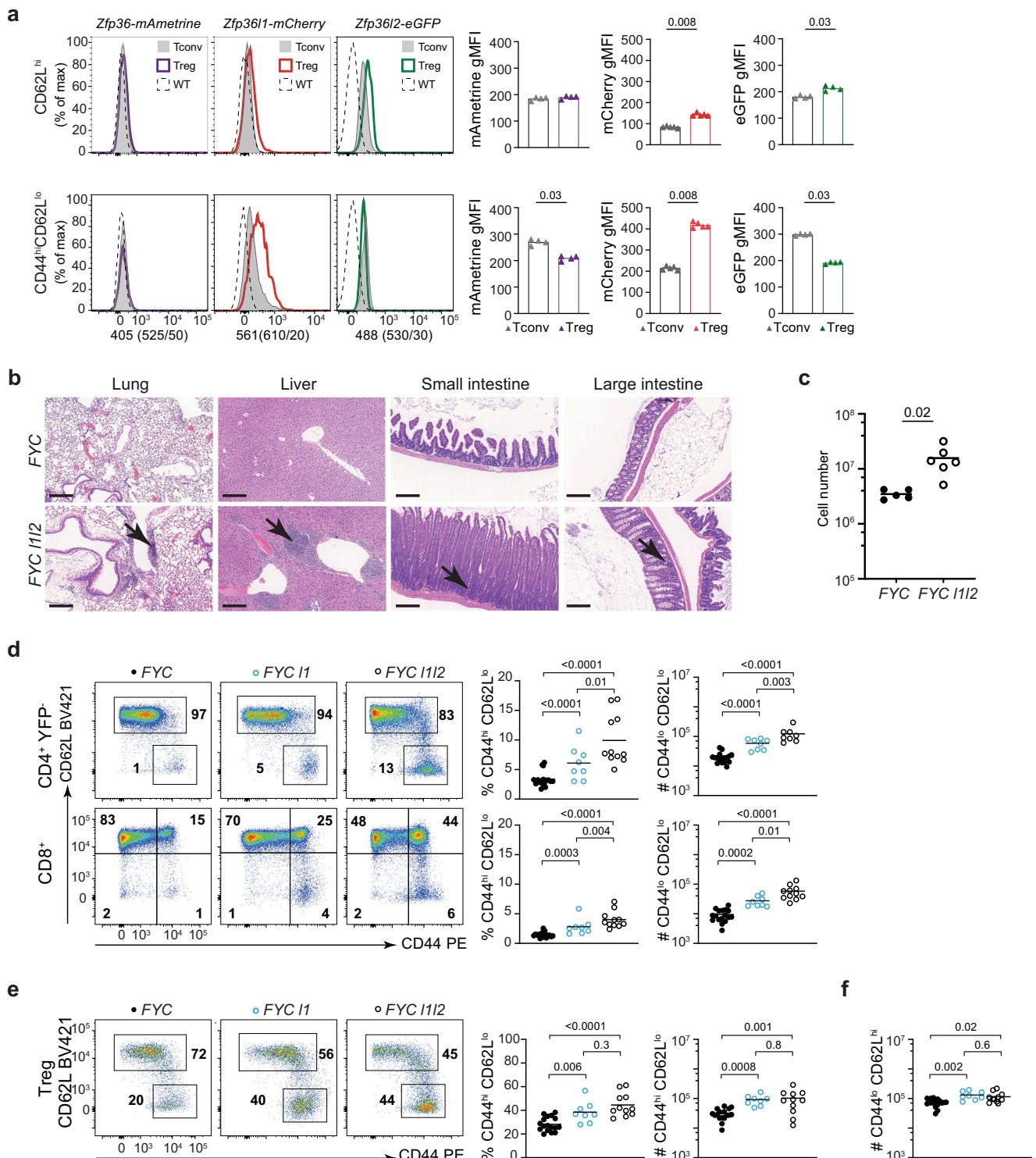

recombination of both conditional alleles (Supplementary Fig. 3e). Furthermore, by intracellular staining for ZFP36L1 in stimulated T cells we observed a seven-fold reduction in the ZFP36L1 protein only in *icre*-positive YFP⁺FOXP3⁺ Treg cells but not in *icre*-negative YFP⁻FOXP3⁺ Treg cells, or in Tconv cells (Supplementary Fig. 3f). Differential expression analysis using DESeq2 revealed 249 genes were increased (Supplementary Data 1) and 340 genes decreased (Supplementary Data 2) in the knockout Treg cells compared to control cells (FDR-adjusted $P$ value < 0.05) (Fig. 2b). The mRNAs encoding *Foxp3*, *Ctla4*, *Ikzf2* (Helios), *Icos*, *Tgfb1* and *Lrrc32*, key genes involved in Treg cell function, were not differently expressed (Fig. 2b).

Individual-nucleotide resolution cross-linking and immunoprecipitation (iCLIP) and its variations[35] are powerful methods for mapping protein RNA interactions. These methods require large amounts of starting material, and it is not yet feasible to obtain high-quality data from rare cells such as Treg cells. Therefore, to identify transcripts that could be bound and thus directly regulated by ZFP36-family members in Treg cells we used previously published data (Supplementary Data 3). To perform the analysis, we classified genes in three categories: genes detected by CLIP in the published data (designated as CLIP); genes whose 3'UTRs contain an AU-rich element (designated as ARE), defined as 2× UAUU separated by up to three nucleotides; and

**Fig. 1 | *Zfp36l1* and *Zfp36l2* are essential in Treg cells to maintain immune homeostasis. a** Representative histogram overlays comparing expression of each reporter in Treg cells (CD4$^+$ CD25$^+$ FR4$^+$) versus Tconv (CD4$^+$ CD25$^-$ FR4$^-$) cells in either the naive (CD44$^{lo}$CD62L$^{hi}$) or effector (CD44$^{hi}$CD62L$^{lo}$) subset, as indicated. Gating as in fig. S1a. In each example, Tconv cells are represented by the gray shaded histogram (and symbol); Treg cells are represented by the bold line: mAmetrine−ZFP36 (purple line and symbol; *n* = 4); mCherry−ZFP36L1 (red line and symbol; *n* = 5) and eGFP−ZFP36L2 (green line and symbol; *n* = 4). Corresponding cells from wild-type mice are indicated by the dashed line. Scatter plots represent the geometric mean fluorescence intensity (gMFI) of each reporter as indicated, with each symbol representing an individual mouse, analyzed in a single experiment. **b** Representative images of hematoxylin and eosin staining of sections of the lung, liver, small and large intestine from three male *FYC* and *FYC l1l2* mice at 11 weeks of age which displayed clinical signs (piloerection, intermittent hunched posture, abdominal distension). Lymphoid infiltrates are arrowed. Scale bar represents 200 µm. **c** Total cell number per LN; FYC (*n* = 5), *FYC l1l2* (*n* = 6); key as

shown. Enumeration of *LN* cells from *FYC l1l2* mice displaying clinical symptoms and age-matched *FYC* control mice aged 8-19 weeks; key as shown. *P* values were determined by Mann−Whitney. **d** Representative flow cytometry (FACS) plots and proportion and cell number of effector cells CD44$^{hi}$CD62L$^{lo}$ in the CD4$^+$ YFP$^-$ (upper panel) and CD8$^+$ subsets (lower panel), in single and double conditional knockout male mice (gated as in Supplementary Fig. 2b). Control *icre*-only *FYC*; *FYC l1*; *FYC l1l2*, key as shown. **e, f** Representative FACS plots, showing percentage and number of CD44$^{hi}$ CD62L$^{lo}$ eTreg (**e**), and **f** numbers of CD44$^{lo}$ CD62L$^{hi}$ nTreg cells in *FYC l1* and *FYC l1l2* male mice (gated as shown in Supplementary Fig. 2b). For (**d**–**f**) each symbol represents an individual mouse with the horizontal line representing the mean: Control *icre*-only *FYC* (*n* = 17); *FYC l1* (*n* = 8); *FYC l1l2* (*n* = 11); values are for LN cells from mice aged 10−14 weeks. Data in (**d**–**f**) from at least two independent experiments. *P* values were determined by Mann−Whitney nonparametric test (**a**, **c**) or one-way ANOVA using multiple comparison (**d**–**f**). Source data are provided as a Source Data file.

genes containing an ARE and detected by CLIP (designated as CLIP-ARE). We found that, CLIP, ARE and CLIP-ARE genes were increased in *FYC$^{/+}$ l1l2* compared to *FYC$^{/+}$* Treg cells (Fig. 2c and Supplementary Fig. 4). This approach identifies candidate target transcripts with high confidence and, those with increased expression are more stable in the absence of ZFP36L1 and ZFP36L2 in *Foxp3$^{YFP-icre}$ l1l2* Treg cells. As both ZFP36L1 and ZFP36L2 regulate translation without affecting mRNA abundance[3–6] transcripts identified by this approach with unchanged expression are candidates to be regulated via translational control.

Gene Set Enrichment Analysis (GSEA) using a custom group of gene sets covering different aspects of T-cell biology (Supplementary Data 4), revealed Endocytosis, T-cell receptor signaling and several genes sets associated with cytokine signaling amongst the top ten positively enriched pathways in *Foxp3$^{YFP-icre}$ l1l2*-deficient Treg cells, including: Decreased in IL-2; Decreased in IL-7; JAK STAT signaling pathway; IL-2 Receptor signaling; and Increased in IFNγ (Fig. 2d). Many of the genes contributing most to the enrichment (present in the leading edge) were transcripts shown to interact directly with ZFP36-family members by CLIP (represented in blue (CLIP-ARE) and orange (CLIP) and listed in Supplementary Data 4). Genes enriched in the endocytosis pathway included 25 in the leading edge that were iCLIP targets, encompassing genes involved in the regulation of protein sorting and membrane trafficking (Fig. 2e and Supplementary Data 5). Within the TCR signaling pathway 40 out of 73 genes in the leading edge were iCLIP targets (Supplementary Data 5 and Fig. 2e), including genes that can promote (*Lat, Cd2*) or attenuate (*Rptor, Dgka, Dok2*) activation via the TCR (Supplementary Fig. 5a). We thus evaluated the protein expression of CD5, which is not a target of the ZFP36 family but an indicator of TCR signaling strength[36,37], and the CLIP target NUR77 (encoded by *Nr4a1*), which is expressed early after TCR stimulation. Expression of these proteins did not differ between YFP$^+$ Treg cells from *FYC$^{/+}$ l1l2* and *FYC$^{/+}$* mice (Supplementary Fig. 5b). Thus, we concluded that in a non-inflammatory environment, TCR signaling was not detectably altered in *Foxp3$^{YFP-icre}$ l1l2*-deficient Treg cells. Using the Immunological Genome Project signatures of common γ-chain family cytokines[38] we found that for genes responding to IL-2 or IL-7 signaling in Treg cells, those that were increased in response to these cytokines were decreased in the transcriptome of *Foxp3$^{YFP-icre}$ l1l2*-deficient Treg cells compared to *Foxp3$^{YFP-icre}$* Treg cells from control *FYC$^{/+}$* mice (Fig. 2f). By contrast, genes that were decreased in response to IL-2 or IL-7 signaling were increased in *Foxp3$^{YFP-icre}$ l1l2*-deficient Treg cells (Fig. 2f). Inspection of the genes increased in response to IFNγ in Treg cells revealed that genes decreased in response to IFNγ were not significantly changed, while genes increased in response to IFNγ were just above the 0.05 threshold for significance as increased in the transcriptome of nTreg cells from *FYC$^{/+}$ l1l2* mice compared to that from control *FYC$^{/+}$* mice (Fig. 2g). The GSEA also identified the Treg cell gene signature and Notch signaling pathway to be just beyond the cut-off

for statistical significance. As noted above, *Foxp3*, *Ctla4*, *Ikzf2* and *Icos*, were not differently expressed; and at the protein level we detected only minor reductions for FOXP3 (1.1-fold), CTLA-4 (1.2-fold) IKZF2 (1.4-fold) and ICOS (1.2-fold) between *Foxp3$^{YFP-icre}$ l1l2* and *Foxp3$^{YFP-icre}$* nTreg cells (Supplementary Fig. 6a). FOXP3, CTLA-4 and IKZF2 also were slightly reduced in eTreg cells (1.1;1.3;1.3-fold respectively), but ICOS showed an increase of 1.3-fold between *Foxp3$^{YFP-icre}$ l1l2* and *Foxp3$^{YFP-icre}$* eTreg cells (Supplementary Fig. 6a). The expression of NOTCH1, a known target of ZFP36L1 and ZFP36L2[31,39], that when overexpressed can impair Treg cell function[40], was not different in *Foxp3$^{YFP-icre}$ l1l2* compared to *Foxp3$^{YFP-icre}$* Treg cells (Supplementary Fig. 6b). The 589 differentially expressed genes in the transcriptome of naive *Foxp3$^{YFP-icre}$ l1l2* Treg cells converge on multiple processes which together could contribute to the lack of fitness and function of *l1l2*-deficient Treg cells.

## Decreased cycling of CTLA-4 in RBP-deficient nTreg cells

As conventional dendritic cells (cDC) are important in maintaining T-cell tolerance[41] and cDC phenotype can be regulated by Treg cells[42], we examined the number of cDC in LN. We had observed that LN cellularity was increased four-fold in *FYC l1l2* mice, however, using a gating strategy (modified from ref. 43) relative to the cDC subsets in *FYC* mice, the number of resident CD172a$^+$ XCR1$^-$ cDC2 was selectively increased (ten-fold) compared to the migratory CD172a$^+$ XCR1$^-$ cDC2 and resident and migratory cDC1 (CD172a$^-$ XCR1$^+$) populations (Fig. 3a, b). CTLA-4 can bind to the costimulatory molecules CD80 and CD86 and can capture these molecules by trans-endocytosis from neighboring antigen-presenting cells (APC) to restrict costimulation via CD28[42]. The selective increase in cDC2 numbers therefore prompted us to examine the expression of CD80 and CD86 on cDC in the LN from male mice. We identified elevated CD80 and CD86 expression only in the resident CD172a$^+$ XCR1$^-$ cDC2 but not in the migratory CD172a$^+$ XCR1$^-$ cDC2 population in *FYC l1l2* mice (Fig. 3c) nor in the cDC1 population (Supplementary Fig. 7). Thus, *FYC l1l2* mice fail to limit the expansion of the cDC2 population of DC which express greater amounts of costimulatory molecules.

As defective CTLA-4 function could lead to increased levels of CD80 and CD86, we examined *Ctla4* transcript abundance and splicing. There was no difference in normalized read counts for *Ctla4* mRNA between YFP$^+$ nTreg cells from female *FYC$^{/+}$ l1l2* and control *FYC$^{/+}$* mice (Supplementary Fig. 8a). Nor was there evidence in *FYC$^{/+}$ l1l2* mice of skipping of exon 3 encoding the CTLA-4 transmembrane domain, which would reduce surface expression of CTLA-4 (Supplementary Fig. 8b). CTLA-4 is predominantly localized within intracellular vesicles, which cycle between the cell surface and intracellular stores, and can be rapidly removed from the surface via clathrin-mediated endocytosis[44]. As GSEA had identified enrichment in the endocytosis pathway, we examined the cycling pool of CTLA-4, which can bind ligand at the

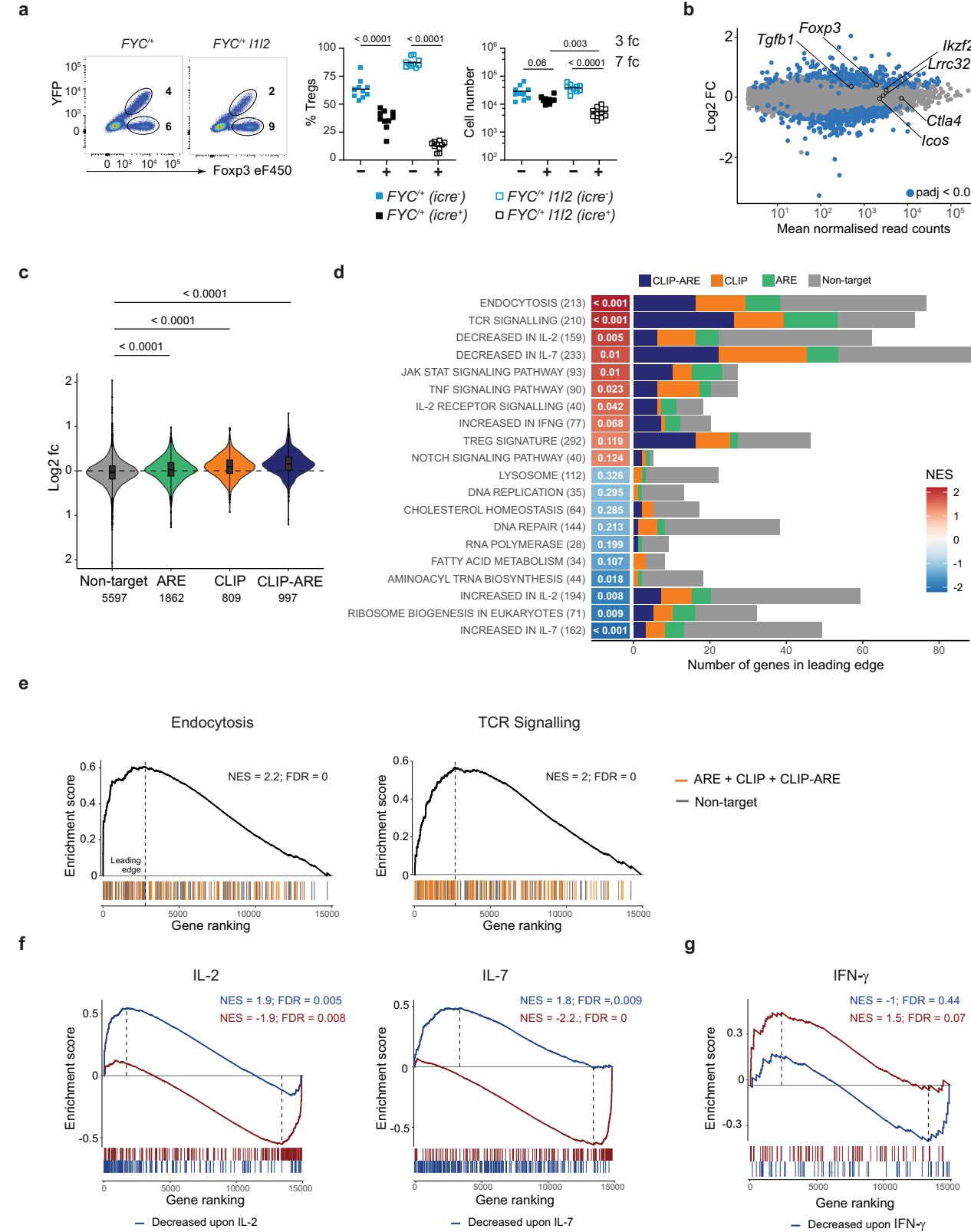

plasma membrane, by incubating unstimulated cells with labeled anti-CTLA-4 antibody at 37 °C for 2 h and comparing this to the total intracellular pool. Constitutive uptake of labeled antibody in vitro visualized in the absence of stimulation, revealed a decrease in the pool of cycling CTLA-4 in YFP+ nTreg cells from male *FYC l1l2* mice compared to nTreg cells from control *FYC* mice, although no difference was observed in

eTreg cells or in the total intracellular pool (Fig. 3d). Consistent with the data from the male mice, in the absence of stimulation, the pool of cycling CTLA-4 in RBP-deficient YFP+ nTreg cells from female *FYC/+ l1l2* mice was also decreased compared to YFP+ nTreg cells from *FYC/+* mice (Fig. 3e) and YFP- nTreg cells in the same mice (Supplementary Fig. 9), but not within the YFP+ eTreg cell subset. These data suggest a cell-

**Fig. 2 | Compromised fitness of RBP-deficient Treg cells. a** Representative FACS plots of FOXP3 and YFP expression in CD4$^+$ cells from female $FYC^{/+}$ and $FYC^{/+}$ $l1l2$ mice; scatter plots showing quantification (% of YFP$^+$ and YFP$^-$ cells within FOXP3$^+$ gate), $n = 10$; key as shown. Each symbol represents data from an individual mouse. Data from at least two independent experiments. $P$ values were determined by one-way ANOVA using multiple comparison. **b** MA plot showing the DESeq2-derived shrunken log$_2$-fold changes in gene expression in $FYC^{/+}$ $l1l2$ compared with $FYC^{/+}$ nTreg cells, padj. <0.05 (two-sided Wald test, implemented by DESeq2) shown in blue, selected genes indicated in black. **c** Violin plot showing the log2-fold change in expression of genes whose 3'UTRs contain an AU-rich element (ARE, defined as 2× UAUU separated by up to 3 nucleotides), ZFP36-family binding (detected by CLIP), or both of these (CLIP-ARE), compared with non-target genes. Only genes with mean normalized read counts >100 were included; the number of genes in each group is indicated; adjusted p values were calculated using ANOVA followed by two-sided Tukey's HSD. For boxplots, center represents the median; bounds represent 25th–75th percentile; whiskers extend 1.5× interquartile range from the upper/lower hinges. Points outside this range are displayed as outliers, although note that some outliers are excluded from the display for the 'Non-target' category. **d** GSEA of custom gene sets (Supplementary Data 4) showing the ten most. positively and negatively enriched pathways in the transcriptomes of Treg cells from $FYC^{/+}$ $l1l2$ compared to $FYC^{/+}$ mice. Genes were ranked based on their expression change upon deletion of $Zfp36l1$ and $l2$ (most significantly increased genes ranked first and most significantly decreased genes ranked last). Numbers in brackets show the total number of genes in the set. The bar graph shows the number of genes in the leading edge, colored using the categories in **c**. The red-blue heatmap shows the Normalized Enrichment Score (NES). Pathways ordered with the highest NES shown at the top; values in white represent the FDR-adjusted $p$ value. **e** GSEA plots for Endocytosis and TCR signaling pathways. Genes whose 3'UTRs contain an ARE and/or ZFP36-family binding (detected by CLIP) are colored in orange. **f, g** GSEA plots using genes with altered expression in the response to the survival factors IL-2 and IL-7 (**f**), and IFNγ (**g**), comparing the genes altered by cytokine stimulation to the transcriptome of Treg cells from $FYC^{/+}$ $l1l2$ mice; genes decreased upon cytokine stimulation shown in blue; genes increased are shown in red. Source data are provided as a Source Data file.

intrinsic role for ZFP36L1 and ZFP36L2 in promoting CTLA-4 cycling in nTreg cells.

## Direct and indirect mechanisms implicated in the regulation of CTLA-4 turnover

To investigate genes regulated by the RBP which might impact on CTLA-4 cycling we scrutinized the leading edge of the endocytosis GSEA (Supplementary Fig. 10). Some genes with known roles in roles in CTLA-4 trafficking were unchanged, including genes encoding TRIM (*Trat1*), LAX, SIT, PLD, TIRC7 (*Tcirg1*), RAB7, RAB8, and DEF6. Similarly, *Usp8* encoding Ubiquitin Specific Peptidase-8, which has a role in CTLA-4 cycling[45], was not differentially expressed. RAB5 and RAB11 have been shown to regulate CTLA-4 trafficking[46] and the transcripts encoding all three paralogs of RAB5, *Rab5a-c*, and *Rab11b* were present in the leading edge of the GSEA. *Lrba*, which encodes a protein that acts upstream of RAB11 and is important for CTLA-4 expression[47], was not in the endocytosis gene set and thus not included in the leading edge but showed a comparable trend to the *Rab5* genes. Of these, *Rab5c* was identified as a direct ZFP36-family target by CLIP, as well as by the presence of an ARE in its 3'UTR (Supplementary Fig. 11), and thus may be regulated directly by the RBP. Other ZFP36-family targets identified by CLIP with a role in CTLA-4 turnover include *Traf6*, encoding an E3 ubiquitin ligase[48], and *Chmp1b* and *Chmp5*, encoding components of the ESCRT-III complex, which are both regulated by USP8[49,50]. The most significantly increased gene in the endocytosis gene set was *Ehd3* which encodes Eps15 homology domain (EHD)-containing protein-3, whose expression was increased approximately threefold in $FYC^{/+}$ $l1l2$ Treg cells; its paralog *Ehd4* was also present in the leading edge and was identified as a direct ZFP36-family target by CLIP, whilst *Ehd3* was not. EHD proteins interact with RAB5, RAB8A and RAB11 and regulate endosomal membrane trafficking[51]. The EHD proteins have been suggested to behave similarly to Dynamin[51], which has a role in CTLA-4 cycling[44]. Another gene with a significant increase in expression, although not directly targeted by the RBP, is *Smap2*. SMAP2 has been proposed to negatively regulate ARF-1[52], a factor required for transport of CTLA-4 to the cell surface[53].

While most of these genes showed only minor changes at the transcript level, which were not on their own significant, these data suggest ZFP36L1 and ZFP36L2 have the potential to regulate CTLA-4 trafficking and turnover at multiple levels. In addition, since several of the genes we identified as candidates for regulating CTLA-4 endocytosis, and/or cycling, were not directly targeted by the ZFP36 family, we examined transcription factors and epigenetic regulators that might be important for controlling expression of genes in the endocytic pathway. Using the ReMap database[54], we identified transcriptional regulators that were frequently bound at the promotors of genes within the endocytosis GSEA leading edge. Taking the genes with the lowest adjusted p values for altered expression in Treg cells from $FYC^{/+}$ $l1l2$ mice, the majority of these showed a trend towards increased expression, and many were also identified as ZFP36-family targets by CLIP, and/or contained an ARE in their 3'UTR (Supplementary Fig. 12a). Among the directly targeted transcription factors were several known to have roles in Treg cells, including *Ets1, Bcl11b, Stat1* (Supplementary Fig. 12b) and *Hopx*[14]. Thus, some endocytosis genes can be regulated by the ZFP36 family through AU-rich elements in their 3'UTRs while targeting of transcriptional regulators provides an indirect mechanism through which endocytosis might be regulated more widely.

## ZFP36L1 and ZFP36L2 are required to maintain IL-2R and IL-7R levels at the cell surface

Based on the finding that CD8 T cells lacking *Zfp36* and *Zfp36l1* were less sensitive to IL-2[28] and from the results of the GSEA we hypothesized that ZFP36L1 and ZFP36L2 promote Treg cell responsiveness to IL-2 and/or IL-7. As both IL-2 and IL-7 induce tyrosine phosphorylation of STAT-5A/B, we examined this in Treg cells ex vivo after rapid fixation of cells following tissue retrieval. In $FYC^{/+}$ $l1l2$ mice we observed a twofold reduction in the frequency of pSTAT5$^+$ cells in FOXP3$^+$YFP$^+$ nTreg cells compared to $FYC^{/+}$ controls, but no difference between the eTreg cell subsets (Fig. 4a), suggesting optimal STAT5 phosphorylation in Treg cells in vivo requires ZFP36L1 and ZFP36L2. As STAT5 can be phosphorylated in response to both IL-2 and IL-7, we measured expression of their receptors. IL-2R is composed of three subunits, IL-2Rα (CD25), IL-2Rβ (CD122) and the common γ chain (CD132). We found no difference in the amount of mRNA encoding the individual subunits between Treg cells from female $FYC^{/+}$ and $FYC^{/+}$ $l1l2$ mice (Fig. 4b). Furthermore, we found no evidence of differential splicing of *Il2ra*, *Il2rb* and *Il2rg* mRNA to favor the expression of soluble forms at the expense of membrane receptors (Supplementary Fig. 13a). However, surface expression of CD25 on YFP$^+$ nTreg cells from the spleen of female $FYC^{/+}$ $l1l2$ mice was reduced two-fold compared to YFP$^+$ nTreg cells from $FYC^{/+}$ mice (Fig. 4c). Among eTreg cells the majority expressed low levels of CD25, but this was reduced 1.5-fold between YFP$^+$ eTreg cells in $FYC^{/+}$ $l1l2$ and $FYC^{/+}$ mice (Fig. 4c). CD122 was also reduced (1.3-fold) in nTreg cells but unchanged in eTreg cells from $FYC^{/+}$ $l1l2$ mice. No difference was observed in expression of CD132 (Fig. 4c). IL-7R is composed of IL-7Ra (CD127) and CD132. We found no difference in mRNA expression of *Il7r* between nTreg cells from female $FYC^{/+}$ and $FYC^{/+}$ $l1l2$ mice (Fig. 4d) or alternative splicing that would favor the secreted form of the IL-7R (Supplementary Fig. 13b). We examined CD127 surface levels and identified that in control mice CD127 was expressed at modestly greater amounts on eTreg compared to nTreg cells (Fig. 4e). CD127 was decreased (1.7- and 1.8-fold,

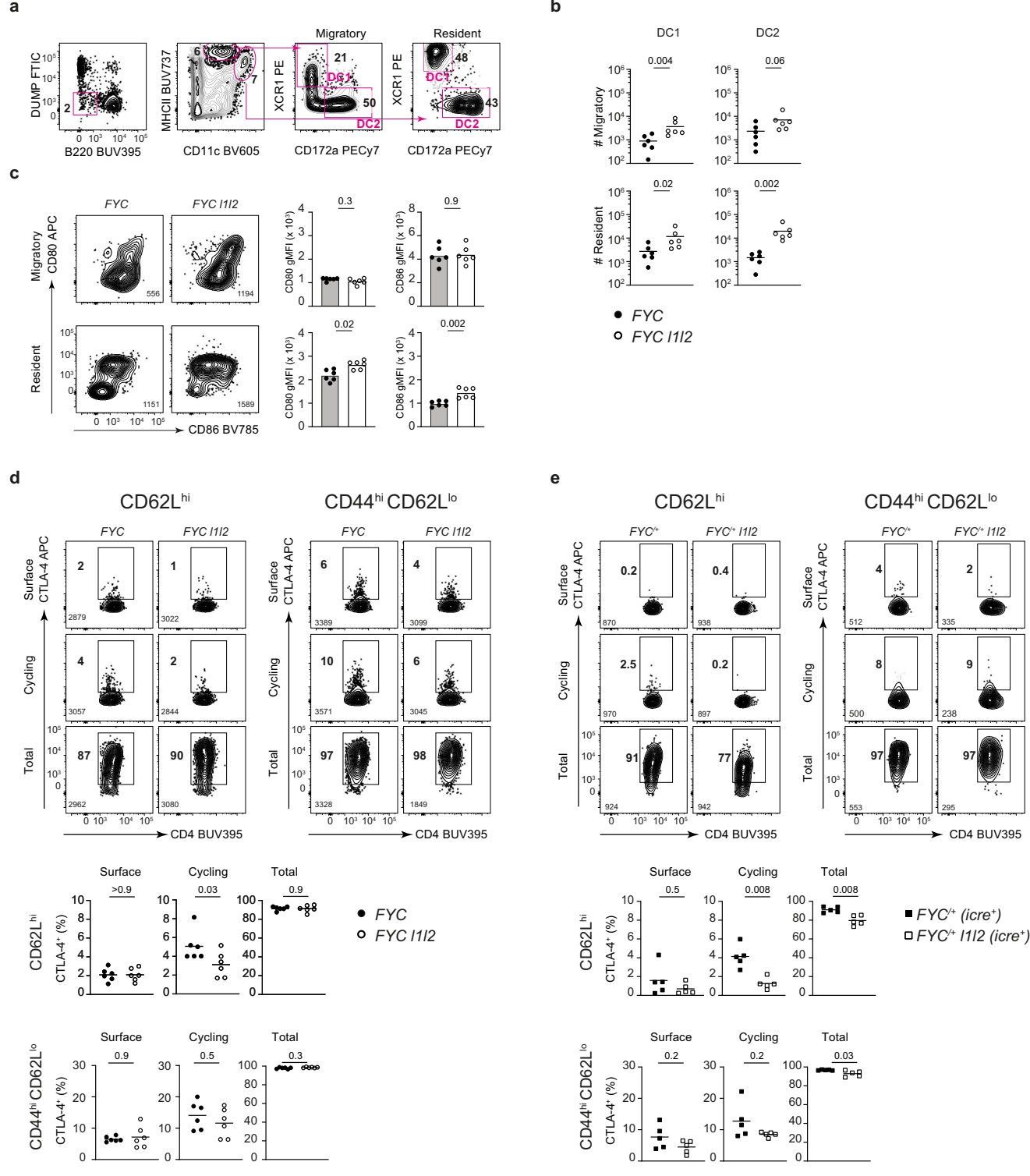

**Fig. 3 | Decreased cycling of CTLA-4 in RBP-deficient nTreg cells. a** Gating strategy for cDC2 (CD172a⁺ XCR1⁻), showing the percentage of events in each gate. Dump channel (FITC: CD3, CD64, F4/80), pre-gated on live, single cells. **b** Enumeration of cells per LN in each DC subset; *n* = 6, key as shown. **c** Representative FACS plots showing CD80 and CD86 expression on CD11c^int MHCII^hi "migratory" (upper panel) and CD11c^hi MHCII^int "resident" (lower panel) cDC2 from spleen from *FYC l1l2* and *FYC* male mice; the number of events in the file

is indicated; *n* = 6, key as shown. **d** Representative FACS plots showing CTLA-4 staining in nTreg cells from spleen from *FYC l1l2* and *FYC* male mice (left panel) and eTreg cells (right panel). Percentage of CTLA-4⁺ Treg cells in each condition (lower panel); *n* = 6. **e** As in (**d**), showing data from *FYC⁺ l1l2 and FYC⁺* female mice; *n* = 5. Each symbol represents data from an individual mouse; key as shown. Data from at least two independent experiments. *P* values were determined by Mann–Whitney test. Source data are provided as a Source Data file.

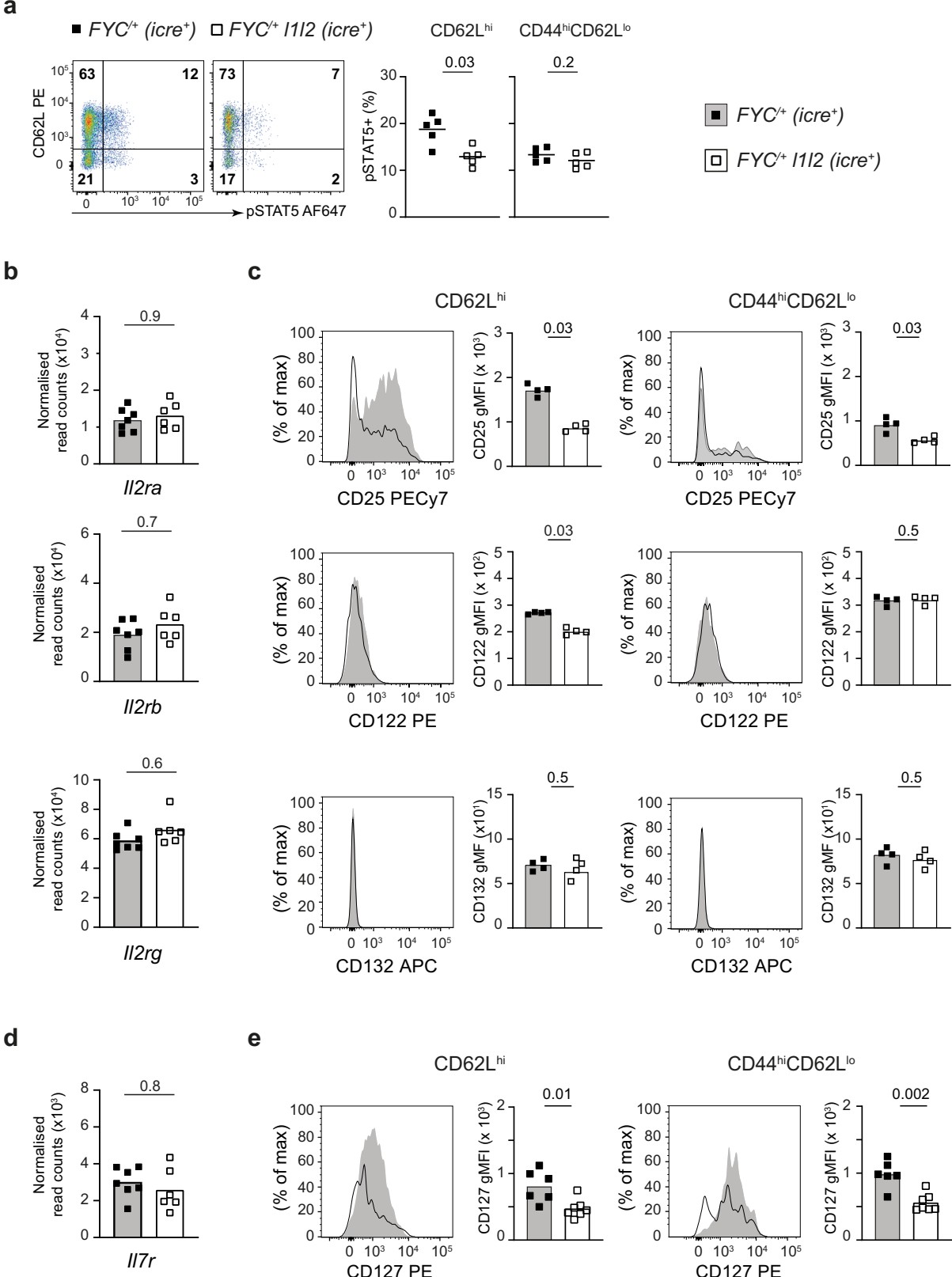

respectively) on YFP⁺ nTreg and eTreg cells from female *FYC^/+ l1l2* mice compared to Treg cells from control *FYC^/+* mice (Fig. 4e). Taken together, these data indicate that ZFP36L1 and ZFP36L2 act in Treg cells to promote the expression of the high-affinity IL-2 and IL-7 receptor in a cell-intrinsic manner.

We analyzed pSTAT5 levels in male mice ex vivo and observed a two-fold reduction in the frequency of pSTAT5⁺ cells in both nTreg and eTreg cells in *FYC l1l2* compared to *FYC* mice (Supplementary Fig. 14). Thus, in the absence of competition with *Zfp36l1/Zfp36l2*-sufficient Treg cells, optimal STAT5 phosphorylation in Treg cells in vivo

**Fig. 4 | ZFP36L1 and ZFP36L2 are required to maintain IL-2R and IL-7R levels at the cell surface. a** Representative FACS plots of YFP⁺ Treg cells from LN from five female mice fixed directly ex vivo and stained for pSTAT5; *FYC⁺*, *FYC⁺ l1l2*; each symbol represents data from an individual mouse; key as shown. **b** Normalized read counts of *Il2ra*, *Il2rb* and *Il2rg* in YFP⁺ nTreg cells from *FYC⁺* (n = 7) and *FYC⁺ l1l2* mice (n = 6); key as shown. Read counts were normalized using size factors derived from the overall DESeq2 analysis of all genes; *P* value determined using two-sided *t* test with FDR correction; key as in (**a**). **c** Representative histogram overlays of CD25, CD122 and CD132 expression on YFP⁺ nTreg cells (left panel) and eTreg cells (right

panel) from the spleen from four female *FYC⁺* and *FYC⁺ l1l2* mice; gMFI of CD25, CD122 and CD132; key as in (**a**). **d** Normalized read counts of *Il7r* in YFP⁺ nTreg cells from *FYC⁺* (n = 7) and *FYC⁺l1l2* mice (n = 6); key as in (**a**). Read counts normalized and *P* value calculated as in (**b**). **e** Representative histogram overlays of CD127 expression on YFP⁺ nTreg cells (left panel) and eTreg cells (right panel) from the spleen from female *FYC⁺* (n = 6) and *FYC⁺ l1l2* mice (n = 7); gMFI of CD127; key as in (**a**). *P* values determined using Mann–Whitney test (**a**, **c**, **e**). Source data are provided as a Source Data file.

requires ZFP36L1 and ZFP36L2. To establish whether IL-2R and IL-7R were affected in *l1l2*-deficient Treg cells from male mice we analyzed their surface expression. We found that expression of CD25 was diminished on nTreg cells from *FYC l1l2* mice (1.3-fold) compared to control *FYC* mice whereas CD122 and CD132 were not reduced (Supplementary Fig 15a). CD127 expression was reduced in both nTreg (1.7-fold) and eTreg cells (two-fold, Supplementary Fig. 15b). Together, these data suggest that ZFP36L1 and ZFP36L2 are key to maintaining CD25 and CD127 receptor levels at the cell surface, independently of the competitive environment.

## ZFP36L1 and ZFP36L2 promote Treg cell sensitivity to IL-2 and IL-7

To analyze if the diminished levels of CD25 and/or CD127 affected responses to IL-2 and IL-7 we measured the in vitro response of Treg cells to a range of doses of IL-2 and IL-7. YFP⁺ nTreg cells from *FYC⁺ l1l2* mice showed diminished responses when cultured with IL-2 at 0.3 and 1 ng/ml but no difference was found in eTreg cells (Fig. 5a). Quantitation of the total amount of STAT5 in YFP⁺ FOXP3⁺ cells in female *FYC⁺ l1l2* and *FYC⁺* mice was not different (Supplementary Fig. 16a). Moreover, *icre*-negative Treg cells from female *FYC⁺ l1l2* and *FYC⁺* mice responded in a similar manner to IL-2 (Supplementary Fig 16b) indicating a cell-intrinsic defect of RBP-deficient Treg cells in their response to IL-2. Furthermore, in *FYC l1l2* male mice the response to IL-2 was also diminished compared to control *FYC* mice (Fig. 5b). Collectively, these data indicate that ZFP36L1 and ZFP36L2 in Treg cells promote an optimal response to IL-2.

Both naive and effector YFP⁺ Treg cells from *FYC⁺ l1l2* mice showed a striking reduction in their pSTAT5 response to IL-7 compared to YFP⁺ Treg cells from *FYC⁺* mice (Fig. 5c). This was also the case for nTreg and eTreg cells from male *FYC l1l2* mice (Fig. 5d). In *FYC⁺* and *FYC⁺ l1l2* females, *icre*-negative Treg cells were more sensitive to IL-7 than *icre*-positive Treg cells from the same mouse. Therefore, expression of the *FYC* allele correlates with a minor reduction in pSTAT5 in response to IL-7 (1.2-fold, Supplementary Fig. 16c). However, the difference between *icre*-positive and *icre*-negative Treg cells in the same animal was greater in *FYC⁺ l1l2* mice compared to control *FYC⁺* mice (Supplementary Fig. 16c), supporting a cell-intrinsic role for ZFP36L1 and ZFP36L2 in the Treg cell response to IL-7. As CD122 and CD132 can form a receptor of lower affinity for IL-2 and still signal in the absence of CD25[55], this may account for the greater defect we observe in the response to IL-7 compared to that for IL-2. These results demonstrate that ZFP36L1 and ZFP36L2 are essential in Treg cells to maintain optimal sensitivity to IL-7 and IL-2.

## ZFP36L1 and ZFP36L2 limit Treg cell sensitivity to IFNγ

As the bulk RNA-seq data from heterozygous female mice suggested a possible enhancement of the IFNγ signaling pathway, we evaluated the response to IFNγ. We used intracellular flow cytometry to detect phosphorylation of STAT1 on Tyrosine-Y701 following stimulation with a range of doses of recombinant mouse IFNγ. We observed an enhanced response to IFNγ in YFP⁺ nTreg cells from *FYC⁺ l1l2* mice compared to YFP⁺ nTreg cells from control *FYC⁺* mice (Fig. 6a). YFP⁺ eTreg cells showed a much lower response than YFP⁺ nTreg cells, which was comparable between genotypes except at the highest

concentration. Significantly, YFP⁺ nTreg cells from *FYC⁺ l1l2* mice showed an enhanced response compared to YFP⁻ nTreg cells from the same mouse indicating the effect was cell-intrinsic (Supplementary Fig. 17a). Quantitation of the total amount of STAT1 in YFP⁺ FOXP3⁺ cells in female *FYC⁺ l1l2* and *FYC⁺* mice confirmed this was not different (Fig. 6b and Supplementary Fig. 17b). In addition, we examined the response to IL-6, which also promotes STAT1 Y701 phosphorylation, and found no increase in STAT1 phosphorylation between YFP⁺ Treg cells from control *FYC⁺* and *FYC⁺ l1l2* mice following in vitro stimulation with a range of doses (Fig. 6c and Supplementary Fig. 17c). Thus, the limiting effects of ZFP36L1 and ZFP36L2 on STAT1 phosphorylation were specific to the IFNγ pathway and indicate these RBP restrain IFNγ signaling in Treg cells in a cell-intrinsic manner.

## Activated phenotype of Treg cells from *FYC l1l2* mice

To explore the heterogeneity of Treg cells and gain further insight into how ZFP36L1 and ZFP36L2 regulate Treg cell function in an inflammatory environment, we performed single-cell sequencing of RNA (scRNA-seq) on Treg cells sorted from the peripheral LN of male *FYC* and *FYC l1l2* mice (Supplementary Fig. 18a). We analyzed high-quality transcriptomes of 5824 cells from *FYC* mice and 4163 cells from *FYC l1l2* mice. Using a graph-based clustering approach Treg cells were grouped into eight subpopulations and represented using a uniform manifold approximation and projection (UMAP) for dimensionality reduction (Fig. 7a). Using gene sets derived from a comparison of naive and effector Treg cell transcriptomes (Supplementary Data 6) we applied the Semi-supervised Category Identification and Assignment (SCINA) algorithm[56] to assign cell type identities. We found that cells in clusters 0, 2 and 5 were frequently classified as nTreg cells, whereas clusters 3, 4 and 7 were primarily comprised of eTreg cells (Fig. 7a, b and Supplementary Fig. 18b). Differential gene expression analysis identified markers for each cluster except for cluster 1, which did not show enrichment for any distinctive genes (Supplementary Data 7). The four most frequently detected genes in each cluster in comparison to every other cluster are shown (Fig. 7c), further illustrating that clusters 0, 2, and 5 exhibited genes typical of nTreg cells (e.g., *Sell*, *S1pr1*). Genes characteristic of eTreg cells (e.g., *Icos*, *Maf*) were not detected or lowly expressed in clusters 0, 2, and 5, but frequently detected in clusters 3, 4 and 7. Cells in cluster 7 expressed markers characteristic of effector and highly proliferative cells, including *Mki67* and histone genes, in addition to a higher number of genes, (Supplementary Fig. 18c) suggestive of a highly proliferative population. Thus, the individual clusters could broadly be distinguished by characteristic marker expression with a distinct distribution corresponding to naive and effector Treg cell subtypes.

The distribution of clusters of Treg cells from *FYC* and *FYC l1l2* mice was not equivalent. Treg cells from *FYC l1l2* mice represented only 20% of cells in cluster 0 and 15% in cluster 2, whereas 80% of cells in cluster 4 were represented by *FYC l1l2* Treg cells (Fig. 7d). As we found that ZFP36L1 and ZFP36L2 limit Treg cell sensitivity to IFNγ, we investigated genes characteristic of the IFNγ response (defined in Supplementary Data 8). We used the SCINA algorithm to assign cell type identities in the scRNA-seq data and found a higher proportion of Treg cells from *FYC l1l2* mice across most clusters associated with an IFNγ gene signature (Fig. 7e). Activity of the IFNγ signaling pathway is necessary for the

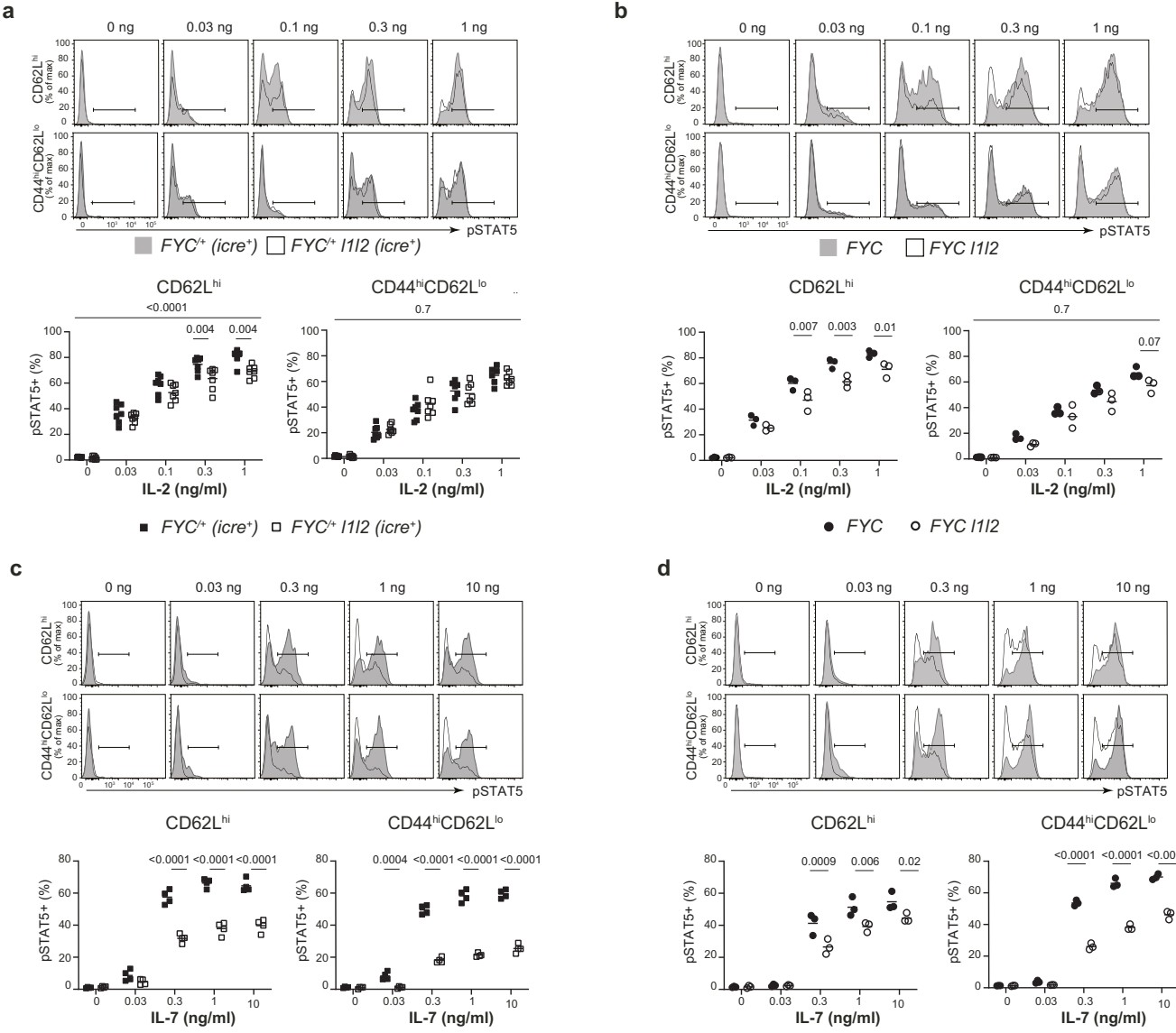

**Fig. 5 | ZFP36L1 and ZFP36L2 promote Treg cell sensitivity to IL-2 and IL-7.**
**a**, **b** Frequency of pSTAT5+ cells in nTreg cells and eTreg cells isolated from the spleen from female $n = 7$ (**a**) or male mice $n = 3$ (**b**) following stimulation for 30 min with a range of concentrations of IL-2; key as shown. **c**, **d** Frequency of pSTAT5+ cells

in nTreg cells and eTreg cells isolated from the spleen from female $n = 4$ (**c**) or male mice $n = 3$ (**d**) following stimulation for 30 min with a range of concentrations of IL-7; key as shown. $P$ values determined using two-way ANOVA with multiple comparison. Source data are provided as a Source Data file.

differentiation of CXCR3+ Treg cells, thus we examined expression of *Cxcr3* and other effector molecules and found enrichment of *Cxcr3* in clusters 3 and 4 (Fig. 7f). In addition, *Gata3* and *Pdcd1* (encoding PD1), which is expressed by Treg cells during activation, were also most frequently detected in cluster 4 (Fig. 7h), which is over-represented in *FYC l1l2* Treg cells. Consistent with these observations, flow cytometry revealed an increase in the proportion (Fig. 7i) and number of CXCR3+ Treg cells (Supplementary Fig. 18d) compared to control mice. GATA3hi Treg cells and YFP+CXCR5+PD1+ T follicular regulatory (Tfr) cells were also increased in proportions (Fig. 7j, k) and numbers (Supplementary Fig. 18d), in *FYC l1l2* mice compared to control mice. Although we found a three-fold reduction in the proportion of RORγt+ Treg cells in *FYC l1l2* mice compared to controls, there was no difference in cell number (Supplementary Fig. 18e), indicating these cells are unable to control the influx of inflammatory cells in the intestine.

Consistent with the increased proportion of CXCR3+ and GATA3+ Treg cells and the decreased proportion of RORγt+ Treg cells in *FYC l1l2* mice, stimulation of splenocytes from male *FYC l1l2* mice revealed

increased frequencies of Treg cells producing IFNγ (six-fold), IL-4 and IL-10 and a reduced proportion producing IL-17 (Fig. 7l). Analysis of the gMFI of these cytokines revealed that Treg cells from *FYC l1l2* mice produce marginally more IFNγ (1.3-fold) and less IL-17 (1.7-fold) compared to Treg cells from *FYC* mice, whilst no difference was found in the production of IL-4 or IL-10 (Supplementary Fig. 18f). FOXP3− CD4+ T cells were also enriched for IFNγ, IL-4 and IL-10-positive cells (Fig. 7l) and produced more IL-10 (Supplementary Fig. 18f). Therefore, the inflammatory environment in *FYC l1l2* mice was not a result of a deficiency of IL-10. Moreover, we found three times more CD8+ T cells producing IFNγ in *FYC l1l2* mice, and the production of IFNγ by these cells also increased (1.6-fold) (Supplementary Fig. 18g). Thus, the expanded CXCR3+ and GATA3+ Treg cell subsets in *FYC l1l2* mice are unable to limit the expansion of activated CD4 and CD8 T cells which produce cytokines that may contribute to the loss of immune homeostasis in *FYC l1l2* mice.

As inflammation can lead to IFNγ and IL-10 production, we also measured the production of these two cytokines in Treg cells from *FYC/+*

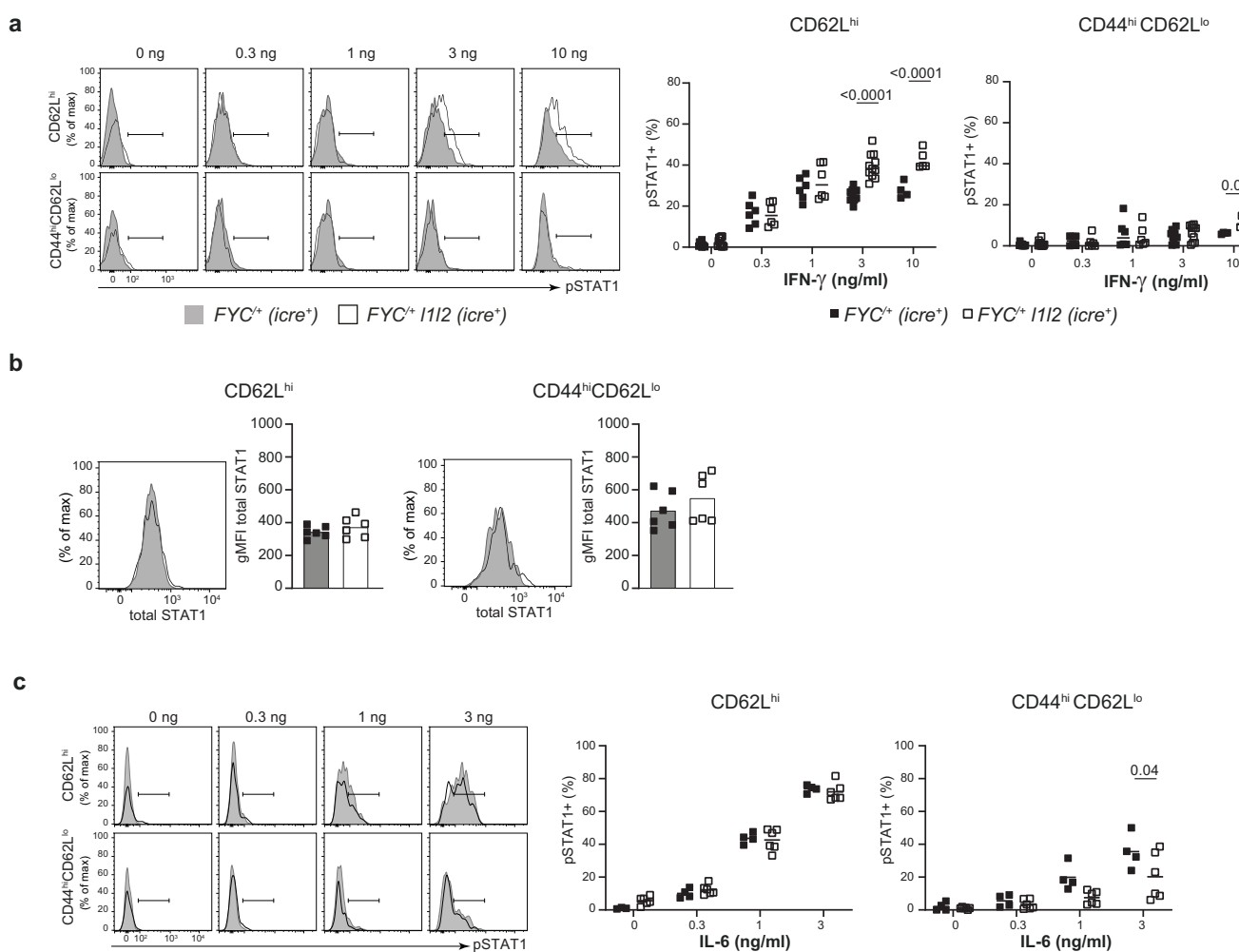

**Fig. 6 | ZFP36L1 and ZFP36L2 limit Treg cell sensitivity to IFNγ. a** Frequency of pSTAT1+ cells detected in YFP+ nTreg cells and YFP+ eTreg cells from female mice following stimulation for 30 min with a range of concentrations of IFNγ. Representative histogram overlays of pSTAT1 expression (left panel) and compiled data (right panel); data from at least four individual *FYC/+* control mice and at least five individual *FYC/+ l1l2* mice; key as shown. **b** FACS analysis of total STAT1 expression in YFP+ Treg cells from six female *FYC/+* and six *FYC/+ l1l2* mice, showing representative histogram overlays of STAT1 expression and quantitation; key as in (**a**). **c** Frequency of pSTAT1+ cells detected in YFP+ nTreg cells and YFP+ eTreg cells from female mice following stimulation for 30 min with a range of concentrations of IL-6. Representative histogram overlays of pSTAT1 expression (left panel) and compiled data (right panel); data from four individual *FYC/+* control mice and six individual *FYC/+ l1l2* mice; key as in (**a**). *P* values determined using two-way ANOVA with multiple comparison (**a**, **c**). Source data are provided as a Source Data file.

*l1l2* mice. Analysis of IFNγ produced by Treg cells in *FYC/+ l1l2* mice revealed a higher proportion of YFP+ Treg cells that were IFNγ+ compared to either YFP+ Treg cells in *FYC/+* mice (three-fold increase*)* or *icre*-negative cells in the same mouse (Supplementary Fig. 19a). In addition, we observed higher amounts (3.6-fold increase) of *Ifng* at the RNA level in the RNA-seq data from YFP+ nTreg cells from *FYC/+ l1l2* mice compared to YFP+ nTreg cells from *FYC/+* mice (Supplementary Fig. 19b). This indicates that the increase in the proportion of IFNγ+ Treg cells observed in *FYC* male mice is not solely caused by inflammation and that these RBP have a cell-intrinsic role in repressing IFNγ production by Treg cells. Because the increase in proportion is smaller in the absence of inflammation when compared to the increase observed in male mice (Fig. 7l), we conclude that the phenotype is exacerbated by the inflammatory environment. We observed no difference in the proportions of IL-10+ cells between YFP+ Treg cells from female *FYC/+ and FYC/+ l1l2* mice (Supplementary Fig. 19c). Thus, the increased proportions of IL-10+ *FYC l1l2* Treg cells are likely a result of the inflammation in these mice.

**IFNγ is a driving force for the expansion of effector T cells**. The enhanced sensitivity of *l1l2*-deficient Treg cells to IFNγ together with the increased number of Treg cells showing an IFNγ signature in the scRNA-seq and the increased proportion of Treg cells producing IFNγ, suggested that ZFP36L1 and ZFP36L2 act in Treg cells to regulate the IFNγ signaling axis. The production of IFNγ by Treg cells and their responsiveness to it are essential for immune homeostasis, and in vivo IFNγ can mediate both pro- and anti-inflammatory processes. Thus, we sought to test the physiological role of IFNγ, while avoiding complications arising from a complete absence of IFNγ, by generating *FYC l1l2* mice heterozygous for *Ifng*. The majority of *Ifng+/− FYC l1l2* mice were healthy, with only one mouse from 23 that developed skin inflammation, compared to ten out of 45 *FYC l1l2* mice developing clinical symptoms which exceeded the severity limit (marked piloerection and intermittent hunched posture, Fig. 8a). *FYC l1l2* mice had elevated levels of IFNγ, TNF, IL-2, IL-6, and IL-10 in the serum compared to *FYC* control mice, however, the amounts of these cytokines were reduced in *Ifng+/− FYC l1l2* compared to *FYC l1l2* mice, with none maintaining a significant increase relative to *FYC* controls (Fig. 8b). The increased number of effector cells in the Treg cell, CD44hi CD62Llo CD4+YFP−, CD44hi CD62Lhi and CD44hi CD62Llo CD8+ subsets in *FYC l1l2* mice was also diminished by loss of one *Ifng* allele but remained two- to three-fold higher than in *FYC* control mice in each subset (Fig. 8c). The number of Tfh, Tfr and GC B cells in *FYC l1l2* mice was significantly

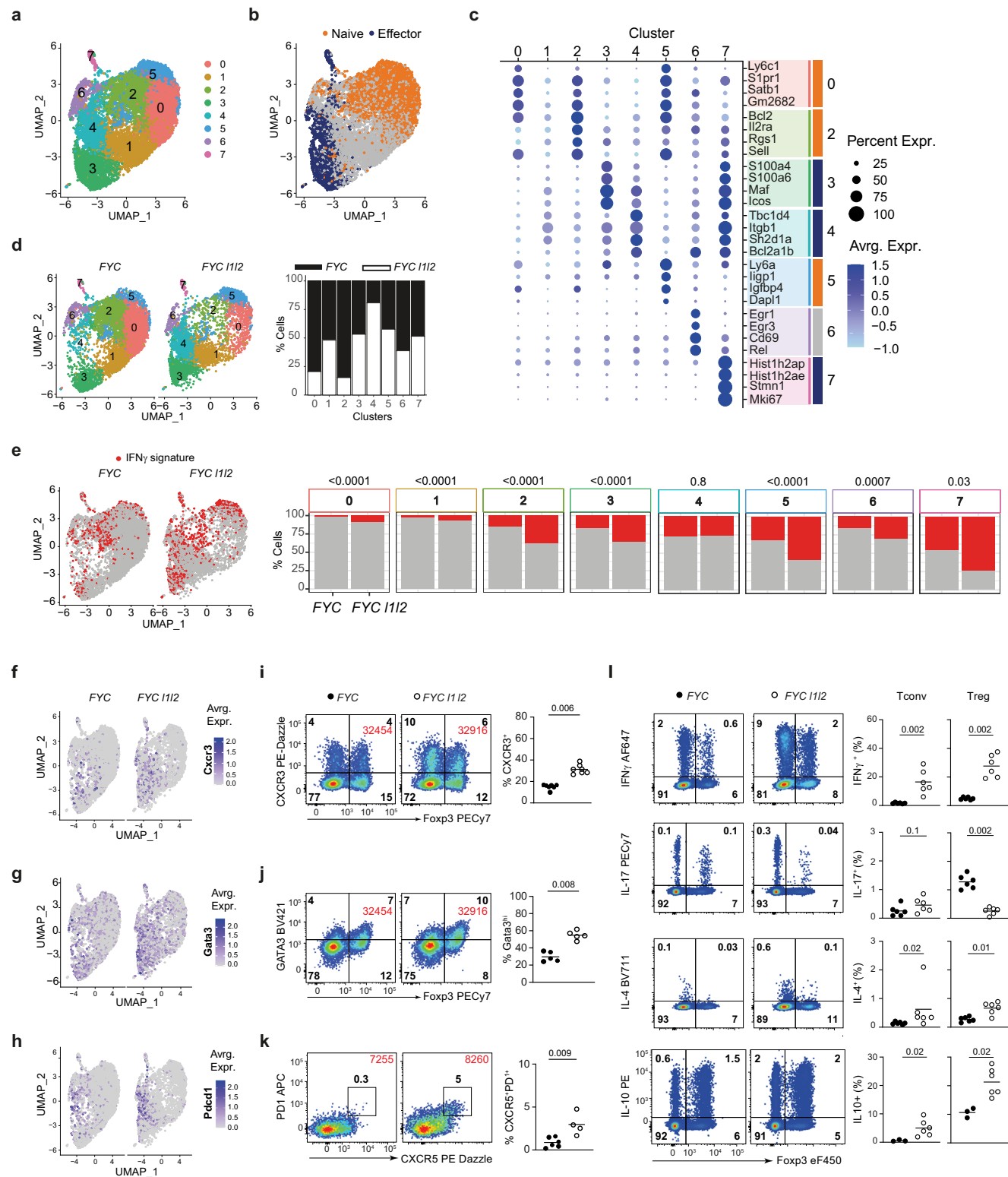

increased in comparison to the *FYC* controls, but these numbers were reduced in *Ifng^+/−^ FYC l1l2* mice, although remained four-fold higher in both Tfh and Tfr subsets (Fig. 8d, e), whilst the population of GC B cells remained ten-fold higher (Fig. 8f). Serum IgG2c and IgE levels were elevated in *FYC l1l2* mice compared to *FYC* controls. In *Ifng^+/−^ FYC l1l2* mice the amount of IgG2c was reduced to that found in control mice (Fig. 8g). However, the concentrations of IgE were still higher (60-fold) in *Ifng^+/−^ FYC l1l2* mice relative to *FYC* control mice, which correlated with the persistence of an elevated number of GC B cells (Fig. 8g).

Thus, these data indicate that IFNγ is a driving force underpinning a complex phenotype in *FYC l1l2* mice.

## ZFP36L1 and ZFP36L2 are required for the maintenance of Treg cell stability

As exposure to IFNγ in the tumor microenvironment can lead to Treg cell instability[26], we analyzed the effect of deleting *Zfp36l1* and *Zfp36l2* on Treg cell stability using a cell-fate mapping system. *Foxp3^eGFP-creERT2^* mice which contain a fusion protein of eGFP with Cre recombinase and

**Fig. 7 | Activated phenotype of Treg cells from *FYC l1l2* mice. a** UMAP representation of scRNA-seq data with cell clusters indicated by color code and numbered 0–7. Single CD4⁺FR4⁺YFP⁺ cells were sorted from apparently healthy male *FYC* and *FYC l1l2* mice. **b** Distribution of nTreg cell and eTreg cell subsets within clusters; key as shown. **c** Dot plot representation showing the percentage of detection (dot size) and scaled expression (dot color) in each cluster for genes more frequently detected and highly expressed in each cluster relative to other clusters (two-sided Wilcoxon rank-sum test; *P* value <0.001). The top four genes for each cluster, based on average log₂ fold change, are shown. Color bar on the right indicates clusters that are primarily comprised of nTreg or eTreg cells; key as in (**b**). **d** UMAP representation of clusters according to genotype as indicated, and percentage contribution of cells to each cluster by genotype; key as shown. **e** UMAP distribution highlighting the location of cells with an expression pattern characteristic of the IFNγ gene signature (indicated in red); and percentage contribution of cells identified as possessing the IFNγ signaling gene signature for each genotype, within

each cluster; adjusted p values from two-sided Chi-square test. Exact *P* values for cluster 0: 2 × 10⁻¹²; 1: 2 × 10⁻⁵; 2: 7 × 10⁻¹⁶; 3: 6 × 10⁻¹⁵; 5: 4 × 10⁻¹⁴. **f–h** UMAP distribution highlighting the location of cells expressing *Cxcr3* (**f**), *Gata3* (**g**), *Pdcd1* (**h**). The intensity of the purple color indicates the scaled expression level for each plot. **i–k** Representative FACS plots gated on CD4⁺ cells and scatter plots showing % of CXCR3⁺ (**i**, *n* = 7), GATA3ʰⁱ (**j**, *n* = 5), and CXCR5⁺ PD1⁺ (Tfr, **k**, *n* = 4) out of all FOXP3⁺ cells; number of events acquired in the CD4⁺ gate is represented in red; key as shown. **l**, IFNγ, IL-17, IL-4, and IL-10 expression in Tconv and Treg cells. Splenocytes were stimulated with PMA/ionomycin for 4 h in the presence of Brefeldin A; FACS plots gated on CD4⁺ cells. Percentage values shown as a % of all CD4⁺ FOXP3⁻ cells (Tconv cells) or CD4⁺FOXP3⁺ cells (Treg cells). Data from six male *FYC* mice (except for IL-10 where *n* = 3), and six *FYC l1l2* mice. Data are representative from at least two independent experiments. *P* values determined using Mann–Whitney test (**i–l**). Source data are provided as a Source Data file.

mutated human estrogen ligand-binding domain (ERT2) inserted into the 3'UTR of the *Foxp3* gene can be used to track FOXP3-expressing cells[57]. As the Cre recombinase only enters the nucleus following tamoxifen exposure, we also introduced a *R26ᶠˡˢᵀᴼᴾ⁻ᵗᵈᴿᶠᴾ* reporter, which consists of a lox-P-flanked transcriptional repressor of a tdRFP transgene[58] so that the cohort of cells that expressed FOXP3 at the time of tamoxifen administration will be heritably marked by RFP expression.

Following administration of tamoxifen-containing diet for 8–9 weeks, we examined the number of *Foxp3ᵉᴳᶠᴾ⁺* Treg cells and *Foxp3ᵉᴳᶠᴾ⁻* ex-Treg cells within the CD4⁺ RFP⁺ subset, the latter indicating cells that had previously expressed FOXP3, but had lost expression at the time of analysis. In *Foxp3ᵉᴳᶠᴾ⁻ᶜʳᵉᴱᴿᵀ² R26ᶠˡˢᵀᴼᴾ⁻ᵗᵈᴿᶠᴾ l1l2* mice, we observed a two-fold increase in the number of ex-Treg cells, indicating that these RBP are required for the maintenance of Treg cell stability (Fig. 9a). We also observed an increase in the proportion of Treg cells producing IFNγ from *Foxp3ᵉᴳᶠᴾ⁻ᶜʳᵉᴱᴿᵀ² R26ᶠˡˢᵀᴼᴾ⁻ᵗᵈᴿᶠᴾ l1l2* mice compared to controls (Fig. 9b). There was no difference between ex-Treg cells from *Foxp3ᵉᴳᶠᴾ⁻ᶜʳᵉᴱᴿᵀ² R26ᶠˡˢᵀᴼᴾ⁻ᵗᵈᴿᶠᴾ l1l2* mice and control mice in terms of IFNγ production following PMA and ionomycin stimulation (Fig. 9b). This was not due to the selection or persistence of non-deleted cells in the ex-Treg cell population as RNA-seq demonstrated efficient deletion in the ex-Treg cells (Supplementary Fig. 21). However, bulk RNA-seq analysis showed a two-fold increase in the amount of *Ifng* mRNA in ex-Treg cells (Fig. 9c) suggesting that ex-Treg cells lacking *l1l2* have the potential to produce more IFNγ in a physiological setting. In the case of TNF, we found no difference at the RNA level but ex-Treg cells lacking *l1l2* produced more TNF (Fig. 9b, d).

To determine if the effect on Treg cell stability was secondary to an inflammatory environment in the *Foxp3ᵉᴳᶠᴾ⁻ᶜʳᵉᴱᴿᵀ² R26ᶠˡˢᵀᴼᴾ⁻ᵗᵈᴿᶠᴾ l1l2* mice, we performed the experiment using females that were heterozygous for the *Foxp3ᵉᴳᶠᴾ⁻ᶜʳᵉᴱᴿᵀ²* which will contain a mixture of eGFP⁺creERT2-expressing and non-expressing cells (eGFP⁻). This enabled us to analyze the effect of deletion of *Zfp36l1* and *Zfp36l2* in Treg cells in the presence of normal Treg cells (Fig. 9e, f). Similar to what was observed in the *FYC* model, in the creERT2 system, *Zfp36l1* and *Zfp36l2*-deficient Treg cells were at a competitive disadvantage. Therefore, a lower number of cre⁺ Treg and ex-Treg cells were found in *Foxp3ᵉᴳᶠᴾ⁻ᶜʳᵉᴱᴿᵀ²/⁺ R26ᶠˡˢᵀᴼᴾ⁻ᵗᵈᴿᶠᴾ l1l2* mice compared to control *Foxp3ᵉᴳᶠᴾ⁻ᶜʳᵉᴱᴿᵀ²/⁺* mice (Fig. 9g). However, the ratio of ex-Treg/Treg cells was higher in *Foxp3ᵉᴳᶠᴾ⁻ᶜʳᵉᴱᴿᵀ²/⁺ R26ᶠˡˢᵀᴼᴾ⁻ᵗᵈᴿᶠᴾ l1l2* mice (Fig. 9h), suggesting that the instability observed in *Zfp36l1* and *Zfp36l2*-deficient Treg cells is cell-intrinsic. These results indicate that ZFP36L1 and ZFP36L2 regulate the availability of IFNγ and are required for the maintenance of Treg cell stability.

## Discussion

All three ZFP36-family paralogues can be expressed by T cells, and, in the absence of infection or inflammation, ZFP36L1 is the most abundant in Treg cells and exerts non-redundant essential functions that are compensated partially by ZFP36L2. ZFP36 proteins can regulate

mRNA at multiple levels by direct binding to AU-rich elements in the 3'UTR of mRNAs. These include promoting RNA degradation, inhibition of translation in the absence of an effect on mRNA decay, and recently, ZFP36L1 was reported to localize mRNAs to endoplasmic reticulum[3–6]. Although the ZFP36 family is a well-established regulator of cytokines, in Treg cells these RBP additionally regulate large numbers of genes that control pathways required by the Treg cells to maintain immune homeostasis.

Treg cells lacking *Zfp36l1 and Zfp36l2* failed to prevent the expansion of cDC2, GC B cells, and effector CD8 T cells. As interactions between cDC2 and Tfh support Tfh priming and GC B responses[59,60], and cDC2 can cross-prime CD8 T cells[61] the increased numbers and elevated expression of CD80 and CD86 on cDC2 could be driving the expansion of activated lymphocytes in *FYC l1l2* mice. The compromised ability of nTreg cells to limit costimulation may arise, in part, from defective CTLA-4 cycling. However, we suggest the role of these RBP in regulating surface receptors is likely to be more extensive and complex. The reduction in surface expression of CD25 and CD127 could result from diminished translation efficiency, altered receptor recycling, or differences in protein turnover. Overall, we identified numerous genes whose expression is altered by deletion of *Zfp36l1/l2* that have the potential to impact on endocytosis and trafficking, including both direct targets of the ZFP36 family, and genes regulated indirectly by the RBP via their regulation of transcription factors.

Interestingly, IL-2 signaling also positively impacts upon Treg cell CTLA-4 function[62] and was impaired in *Zfp36l1* and *Zfp36l2*-deficient Treg cells and this may contribute to the diminished function of CTLA-4. Furthermore, IL-2 sensitivity is diminished in CD8 T cells lacking *Zfp36l1* suggesting ZFP36L1 connects antigen affinity to IL-2 responsiveness[28].

Although IL-2 signaling is essential for Treg cell survival and function[63,64], when deprived of this cytokine IL-7 can sustain Treg cells and contribute to Treg cell homeostasis[64,65]. Mice with conditional deletion of *Il7ra* in Treg cells did not develop autoimmune disease, but IL-7R on Treg cells was important to maintain allograft tolerance in a Treg cell transfer model[66] and IL-7 promotes eTreg cell survival in the skin[67]. In Treg cells we found that ZFP36L1 and ZFP36L2 promote sensitivity to both IL-2 and IL-7. This may account for the reduction in number of *icre*-expressing Treg cells in female mice heterozygous for *icre*, where these cells are in competition for these cytokines. As the development of Tfr has been reported to be inhibited by IL-2[68,69], a defect in IL-2 signaling would be accompanied by the expansion of Tfr as we observe in *FYC l1l2* mice. The accumulation of Tfh and GC B cells may also be augmented by the increased availability of IFNγ[70].

*Zfp36l1* and *Zfp36l2* also specifically limit Treg cell sensitivity to IFNγ. Whilst Th1-like Treg cells can be potent suppressors of the activity, proliferation, and memory formation of CD8 effector T cells[27] and IFNγ-mediated STAT1 activation in Treg cells can promote Treg cell function in alloantigen tolerized mice[25], IFNγ can also promote Treg cell fragility

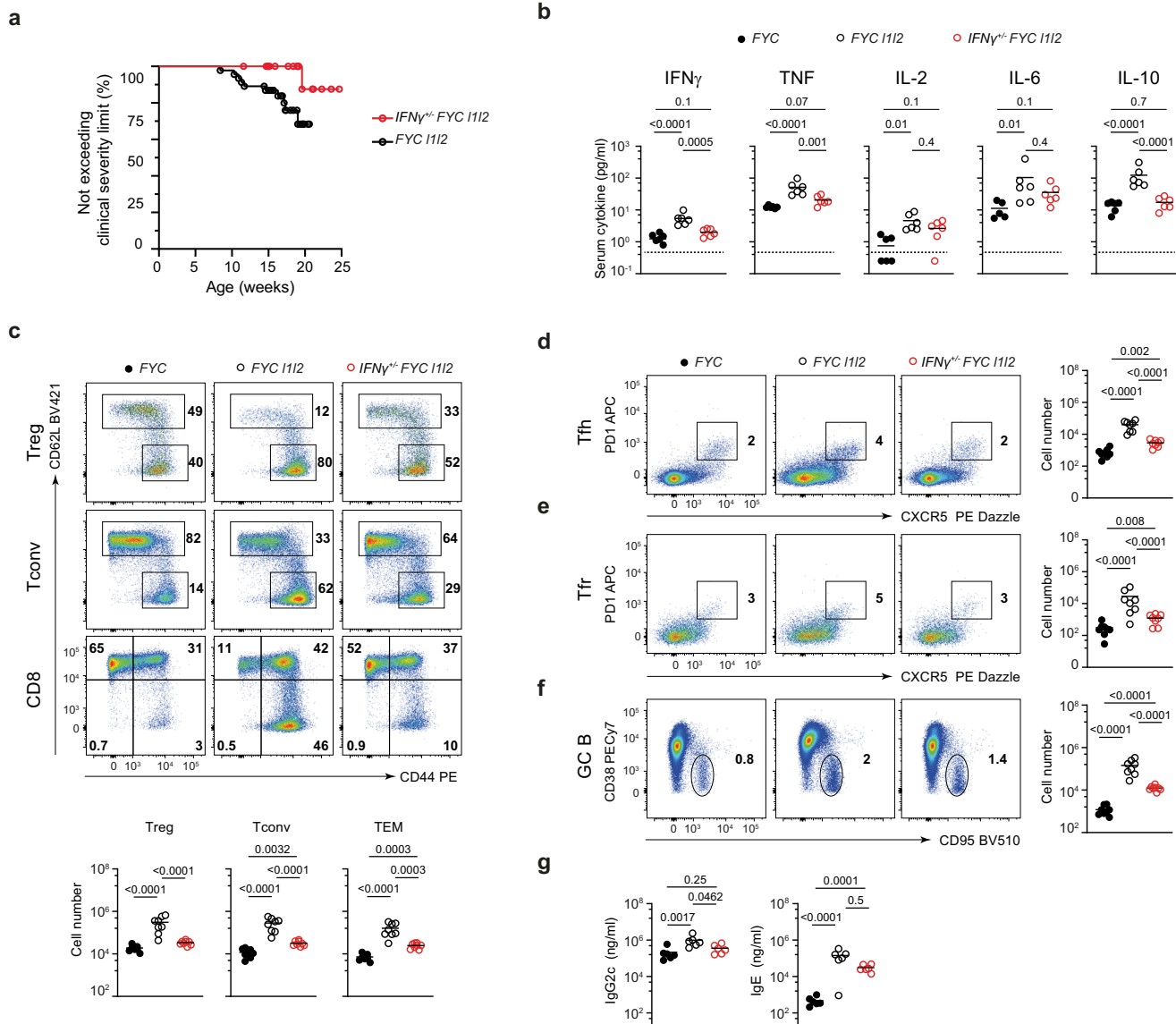

**Fig. 8 | IFNγ is a driving force for the expansion of effector T cells.**
**a** Development of clinical symptoms with age for *FYC l1l2* (*n* = 46) and *Ifng*[+/−] *FYC l1l2* male mice (*n* = 24); key as indicated. Logrank test *P* = 0.03. **b** Serum cytokine levels in *FYC*, *FYC l1l2* and *Ifng*[+/−] *FYC l1l2* male mice; *n* = 6 key as shown. Comparative values for *FYC l1l2* and *FYC*: IFNγ (four-fold); TNF (four-fold); IL-2 (three-fold); IL-6 (nine-fold); IL-10 (nine-fold). Dashed line represents limits of detection.
**c** Representative FACS plots and gating strategy (upper) and enumeration (lower panel) of effector cells in the Treg, Tconv (CD44 hiCD62Llo) and CD8+ TEM (CD44hiCD62Llo) cell subsets. Numbers are shown per single LN in mice aged 15 weeks (*n* = 8). Key as in (**b**). **d**–**f** Representative FACS plots and enumeration of

**d** T follicular helper (Tfh) YFP-CXCR5+PD1+), **e** Tfr (YFP+CXCR5+PD1+), and **f** GC B cells (B220+CD38loCD95+; gated as in Supplementary Fig. 20); *n* = 8. Key as in (**b**). Comparative values for *FYC l1l2* and *FYC*: Tfh (60-fold), Tfr (90-fold), GC B (100-fold). **g** Total serum Immunoglobulin levels quantified by ELISA; *n* = 6. Key as in (**b**). Comparative values for *FYC l1l2* with *FYC*: IgG2c (five-fold), IgE (300-fold). Analyses combine data from two or more independent experiments. **b**–**g** *P* values determined using one-way ANOVA with multiple comparison (**b**–**g**). The data points shown in parts (**b**–**g**) are from individual mice which did not display clinical symptoms. Source data are provided as a Source Data file.

or lineage instability in the tumor microenvironment[26]. We have shown that ZFP36L1 and ZFP36L2 maintain Treg cell stability and that in a non-inflammatory environment ZFP36L1 is increased in the eTreg cell population compared to nTreg cells. It is possible both paralogs increase their expression even more in inflammatory environments to prevent Treg cell instability.

The ability of ZFP36-family members to repress IFNγ expression did not manifest as increased amounts of IFNγ produced by ex vivo stimulated Treg cells but was apparent as a greater frequency of IFNγ producing Treg cells. The capacity for ZFP36L1 and ZFP36L2 in Treg cells to interconnect the IL-2, IFNγ and CTLA-4 pathways and enforce their function, whilst dampening their function in effector CD4 and

CD8 T cells may be relevant to the association of the genes encoding this family of RBP with autoimmune diseases[2].

## Limitations of the study
These phenotypes were not evident in mice with deletion of ZFP36-family proteins using *Cd4*-cre which, taken together with the specific deletion, argues that the phenotypes arise from the function of ZFP36L1 and ZFP36L2 in Treg cells. We have focussed on Treg cells in the absence of intentional immune challenge, and it is unclear how Treg cells deficient in these RBP respond to activation and how this affects transcriptomes, cell function, and the physiology of the mice. We can only speculate as to why 9–12% mice develop

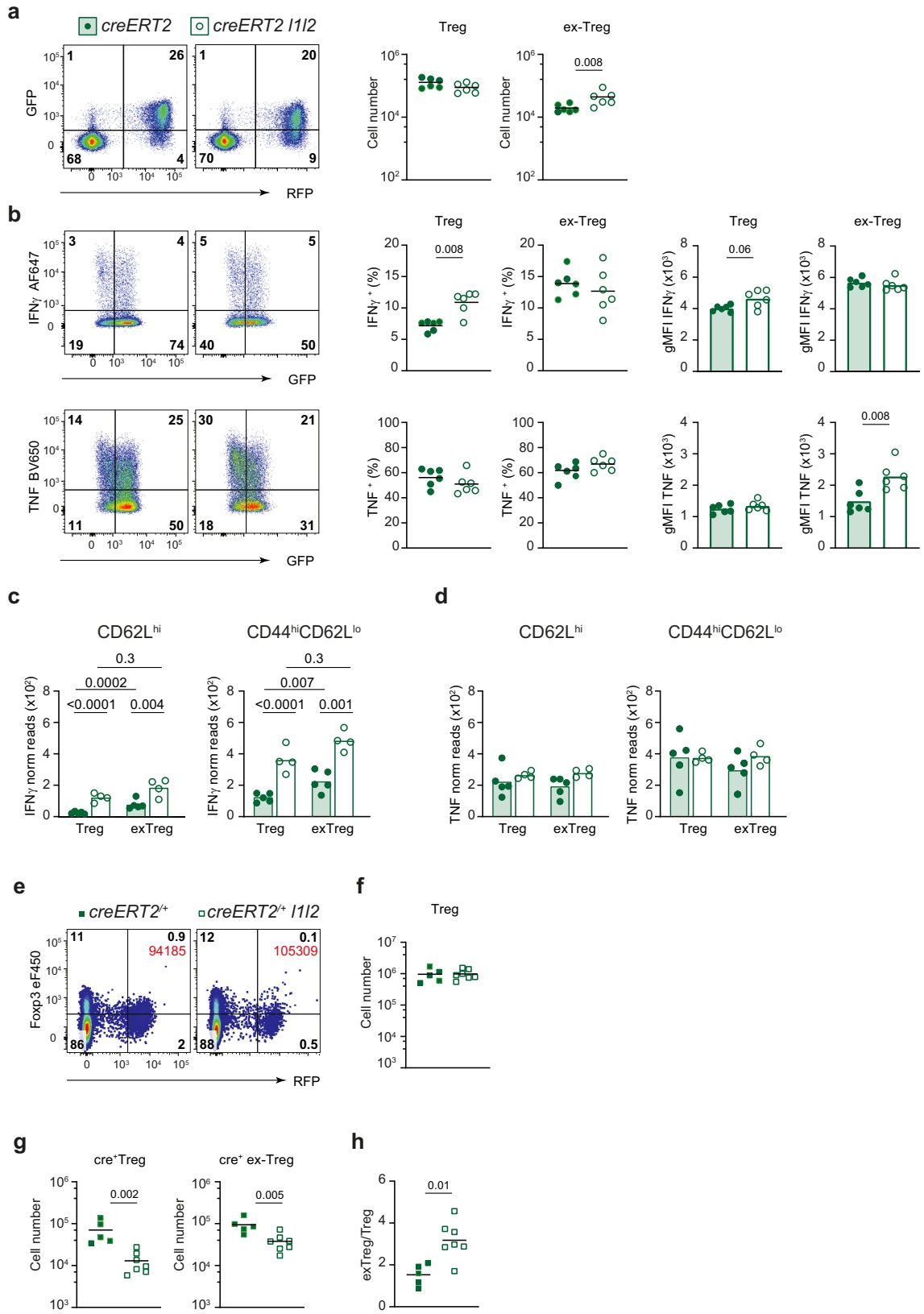

symptoms that exceed the severity limit, but consider potential effects from the local cage environment and the aggressive interactions between male mice that are cohoused, inducing immune responses which may trigger clinical signs in some instances. The compendium of iCLIP targets is from CD4 and CD8 T cells, it is possible some mRNAs unique to Treg cells were not detected. In addition to regulating RNA stability ZFP36L1 and ZFP36L2 can also repress translation[3,4] and may regulate protein localization, thus the use of sensitive proteomics methods could reveal additional direct and indirect targets regulated by the RBP. The detailed mechanism

**Fig. 9 | ZFP36L1 and L2 are required for the maintenance of Treg cell stability.** **a** Representative FACS plots of CD4⁺ T cells from LN from *Foxp3*^eGFP-creERT2^ *R26*^flSTOP-tdRFP^ and *Foxp3*^eGFP-creERT2^ *R26*^flSTOP-tdRFP^ *l1l2* mice fed tamoxifen for 8 weeks, and enumeration of RFP⁺GFP⁺ Treg cells and RFP⁺GFP⁻ "ex-Treg cells"; control, filled symbols; *Foxp3*^eGFP-creERT2^ *R26*^flSTOP-tdRFP^ *l1l2*, open symbols. Values on the flow plot represent percentages of cells in each quadrant; key as shown. **b** Representative FACS plots and quantitation of IFNγ or TNF production by Treg cells or ex-Treg cells from LN following 4 h stimulation in vitro with PMA/ionomycin. Numbers represent percentages of cells in each quadrant. All cells were pre-gated on CD4⁺RFP⁺ (as in (**a**)). Percentages of cells that were positive for IFNγ⁺ or TNF⁺ are from all GFP⁺ cells (Treg cells) or from all GFP⁻ cells (ex-Treg cells). The gMFI represents the values from only the cytokine-positive cells in each subset. Data representative of two independent experiments; each symbol represents data from an individual mouse, *n* = 6.

**c**, **d** Normalized read counts from bulk RNA-seq data for IFNγ (**c**), and TNF (**d**) in Treg cells (sorted as CD4⁺RFP⁺GFP⁺) and ex-Treg cells (sorted as CD4⁺RFP⁺GFP⁻). Cells were further sorted into naive (CD62L^hi^) and effectors (CD44^hi^CD62L^lo^). Key as in a. Both hemizygous *Foxp3*^eGFP-creERT2^ males and homozygous *Foxp3*^eGFP-creERT2^ females were used for data shown in (**a**–**d**). **e**–**h** Representative FACS plots of CD4⁺ T cells from the spleen from five control heterozygous *Foxp3*^eGFP-creERT2/+^ *R26*^flSTOP-tdRFP^ and seven heterozygous *Foxp3*^eGFP-creERT2/+^ *R26*^flSTOP-tdRFP^ *l1l2* female mice fed tamoxifen for 8 weeks. Values represent the percentage of cells in each quadrant and those shown in red represent the number of events acquired in the CD4⁺ gate (**e**); enumeration of all FOXP3⁺ Treg cells (**f**); numbers of cre⁺ Treg cells (FOXP3⁺RFP⁺) and cre⁺ ex-Treg cells (FOXP3⁻ RFP⁺, **g**); ratio of RFP⁺ ex-Treg:Treg cells in spleen (**h**); key as in (**e**). *P* values were determined by Mann–Whitney (**a**, **b**, **f**, **g**, **h**) and one-way ANOVA using multiple comparison (**c**, **d**). Source data are provided as a Source Data file.

of how the RBP regulate CTLA-4 and cytokine signaling in Treg cells remains to be elucidated.

## Methods

All mouse experimentation was approved by the Babraham Institute Animal Welfare and Ethical Review Body. Animal husbandry and experimentation complied with European Union and United Kingdom Home Office legislation.

### Mice

Mice were engineered to express the fluorescent proteins mAmetrine or mCherry upstream and in frame with the start codons of *Zfp36* or *Zfp36l1*, respectively[28]. The targeting vector for *Zfp36l2* was designed to encode eGFP upstream and in frame with the ATG translation start site of ZFP36L2 (Cyagen, Santa Clara, CA, USA). The reporter mice were healthy and fertile (and used for breeding up to 40 weeks of age). *Zfp36*^fl/fl^ (*Zfp36*^tm1Tnr^), *Zfp36l1*^fl/fl^ (*Zfp36l1*^tm1.1Tnr^) and *Zfp36l2*^fl/fl^ (*Zfp36l2*^tm1.1Tnr^) mice have been described previously[31,71] and were maintained on a C57BL/6J background. The following alleles on the C57BL/6J background (obtained from the Jackson Laboratory) were also used: B6.129(Cg)-*Foxp3*^tm4(YFP/icre)Ayr^/J; Jax stock #016959); and B6.129S7-*IFNg*^tm1Ts^ (Jax stock #002287). Animals containing the allele for the tamoxifen-inducible cre *Foxp3*^tm9(EGFP/cre/ERT2)Ayr^/J[57] (Jax stock #016961) and the B6.Cg-*Gt(ROSA)26Sor*^tm1Hjf^/J (Rosa-tdRFP)[58] reporter (Jax stock #038164) were used to fate map Treg cells that had expressed cre (controls) or were crossed to generate mice carrying floxed *Zfp36l1* and *Zfp36l2* alleles. Mice were homozygous for the *R26*^flSTOP-tdRFP^ allele and both hemizygous *Foxp3*^eGFP-creERT2^ males and cre-homozygous females were used in Fig. 9a, b. For the experiments shown in Fig. 8, the cohort of *FYC l1l2* and *Ifng*^+/-^ *FYC l1l2* mice that were analyzed had not been backcrossed to C57BL/6 for seven generations. Any animals which displayed the limiting clinical signs of marked piloerection and intermittent hunched posture, and/or abdominal distension were humanely killed. $CO_2$ inhalation followed by destruction of the brain as confirmation, or cervical dislocation followed by the destruction of the brain were used for humane killing of animals.

For induction of Cre-ERT2–mediated recombination, mice were fed baseline non-tamoxifen-containing diet for 1 week before introduction of the tamoxifen-containing diet *ad libitum* (TD.55125.I, Tamoxifen citrate 400 mg/kg diet, Envigo) for 8–9 weeks.

Mice were bred and maintained in the Babraham Institute Biological Support Unit. Age and sex-matched animals were used as indicated. No primary pathogens or additional agents listed in the FELASA recommendations have been confirmed during health monitoring surveys of the stock holding rooms. Ambient temperature was -19–21 °C and relative humidity 52%. Lighting was provided on a 12-h light: 12-h dark cycle, including 15 min "dawn" and "dusk" periods of subdued lighting. After weaning, mice were transferred to individually ventilated cages with 1–5 mice per cage. Mice were fed CRM (P) VP diet (Special Diet Services) ad libitum and received seeds (e.g., sunflower, millet) at the time of cage cleaning as part of their environmental enrichment.

### Flow cytometry

Single-cell suspensions were prepared from the spleen, peripheral lymph nodes (LN, axillary, brachial, cervical and inguinal) and mesenteric LN (mLN), with antibody staining for surface markers performed essentially as described previously[10] including Fc block (2.4G2) in PBS/2% FCS/2 mM EDTA buffer. Dead cells were excluded using fixable viability dye eFluor780 (65-0865-14, eBioscience), washed, then stained with the antibody cocktail at 4 °C. A full list of antibodies used is provided in Supplementary Data 9. An antibody against GFP (clone FM264G, BioLegend) was used for intracellular detection of YFP which is fused to cre in *Foxp3*^YFP-icre^ mice. A polyclonal antibody against RFP (Rockland) was used for intracellular detection of RFP in *R26*^flSTOP-tdRFP^ reporter mice. For detection of phospho-antibodies, the cells were fixed in neutral buffered formalin (Sigma) diluted in PBS to 2% for 30 min at room temperature, pelleted by centrifugation and cells were resuspended in ice-cold 90% methanol, incubated for 30 min on ice, washed twice, then incubated with the antibody cocktail containing 2.4G2 antibody overnight at 4 °C. For dendritic cell isolation, LN tissue was finely minced then digested with collagenase P (11213865001, Merck, 1 mg/ml in RPMI 2% FCS) for 40 min at 37 °C with agitation, then EDTA added to 5 mM, clumps dispersed using pipetting and the cell suspension filtered. Single-cell suspensions were washed and stained as described.

For ex vivo analysis of pSTAT5 LN cell suspensions were prepared directly into fixation buffer from eBioscience™ Foxp3/Transcription Factor Staining Buffer Set (00-5523-00). For intracellular cytokine detection, LN cells were stimulated for 4 h at 37 °C with 10 ng/ml Phorbol-12-myristate-13-acetate (PMA; 524400, Merck) and 500 ng/ml ionomycin (407952, Merck) in the presence of 5 µg/ml Brefeldin A (420601, Biolegend) in complete IMDM medium (31980-022, Invitrogen; containing 10% FBS, 2 mM Glutamax, 25 mM HEPES, 50 µM 2-mercaptoethanol). Cells were fixed with 2% PFA for 30 min at room temperature, washed twice with Foxp3 permeabilization buffer (eBioscience), and incubated with the antibody cocktail containing 2.4G2 antibody overnight at 4 °C. For transcription factor staining, Foxp3 fixation buffer was used according to the manufacturer's instructions. The acquisition was performed on a Fortessa flow cytometer equipped with 355 nm, 405 nm, 488 nm, 561 nm, and 640 nm lasers (Beckton Dickinson), and the data were analyzed using FlowJo software (v10).

### In vitro cytokine stimulation

Splenocytes were cultured for 30 min at 37 °C with a range of doses of recombinant mouse cytokines (all from Peprotech) IFNγ (# 315-05) IL-6 (# 216-12), IL-2 (# 212-12) IL-7 (# 217-17) in the presence of eF780 viability dye and 150 µM TAPI-0 (Tocris # 5523) in complete IMDM medium.

### Ex vivo CTLA-4 cycling assay

Following red blood cell lysis with Tris-ammonium chloride, CD4⁺ T cells were isolated from the spleen by depletion of CD8⁺, CD11b⁺

CD11c[+], MHCII[+], and B220[+] cells using biotinylated antibodies and M-280 Streptavidin Dynabeads. CD4[+] T cells were cultured for 2 h at 37 °C in the presence of TAPI-0 (100 µM) and 2 ng/ml IL-2. APC conjugated anti-CD152 (CTLA-4) antibody (clone UC10-4F10-11) was added to the cells to detect cycling CTLA-4. Surface CTLA-4 was detected by labeling cells with APC conjugated anti-CD152 for 30 min at 4 °C, and total CTLA-4 was detected after fixation and permeabilization with antibody incubation overnight at 4 °C.

## Histology

For histology, tissue samples were collected into 10% neutral buffered formalin (Sigma). Preparation of slides, H&E staining, pathology and interpretation was performed by Abbey Veterinary Services UK.

## Measurement of serum cytokines and immunoglobulin

Serum was collected from mice by cardiac puncture. Serum cytokines were measured using the MSD pro-inflammatory panel 1. IgG2c and IgE were measured by ELISA using paired antibody sets (IgG2c, Southern Biotech and IgE, 555428, BD Biosciences) according to the manufacturer's protocol.

## Cell sorting for RNA-seq

Single-cell suspensions were prepared from pooled spleen and peripheral LN (inguinal, brachial, axillary, and cervical) from six *FYC*[/+] *l1l2* and seven *FYC*[/+] mice at 7–14 weeks of age. CD4[+] cells were enriched by negative depletion using biotinylated anti-B220, anti-CD8 and anti-CD11b followed by Streptavidin Dynabeads (Dynal). Naive Treg cells were sorted as CD4[+]CD25[+] CD62L[+] YFP[+] cells. All samples were sorted to >95% purity.

Single-cell suspensions were prepared from pooled spleen and LN from four *Foxp3*[eGFP-creERT2] *R26*[flSTOP-tdRFP] *l1l2* mice and five *Foxp3*[eGFP-creERT2] *R26*[flSTOP-tdRFP] control mice (age and sex-matched) following nine weeks of tamoxifen administration. CD4[+] cells were enriched by negative depletion as above. Cells were sorted as: CD4[+]RFP[+]GFP[+]CD62L[hi] (naive Treg cells); CD4[+]RFP[+]GFP[+]CD44[hi]CD62L[lo] (effector Treg cells); CD4[+]RFP[+]GFP[-]CD62L[hi] (naive ex-Treg cells) and CD4[+]RFP[+]GFP[-]CD44[hi]CD62L[lo] (effector ex-Treg cells). All samples were sorted to >95% purity.

For RNA-seq libraries used to identify markers of naive and effector Treg cells, cells were prepared as above from male *Zfp36l1*[fl/fl] *l2*[fl/fl] control mice, aged 9–12 weeks, but with naive Treg cells sorted as CD4[+]CD25[+]CD62L[+]CD44[-] and effector Treg cells as CD4[+]CD25[+] CD62L[-]CD44[+] cells.

To sort Treg cells for single-cell RNA-seq, single-cell suspensions were prepared from pooled LN from one mouse of each genotype at 14 weeks of age. CD4[+] cells were enriched by negative depletion using biotinylated anti-B220, anti-CD8 and anti-F4/80 followed by Streptavidin Dynabeads (Dynal). Treg cells were sorted as CD4[+]FR4[+]YFP[+] cells. All samples were sorted to >95% purity.

## Bulk RNA-seq

RNA was prepared from sorted Treg cells using the RNeasy Micro kit (74004, Qiagen) as described[72], and quality was assessed using the Bioanalyzer RNA chip. cDNA was generated using SMART seq v4 low input RNA kit (634888, Takara). RNA-seq libraries were prepared from cDNA using the Nextera XT kit (FC-131-1096, Illumina). Libraries were sequenced using a 50 bp single-end RapidRun on the Illumina HiSeq2500.

## ScRNA-seq

Cells were labeled using Totalseq oligo-conjugated antibodies (BioLegend) to enable multiplexing of control (hashtag 1: ACCCACCAGTAAGAC) and conditional knockout (hashtag 2: GGTCGAGAGCATTCA) samples, and feature barcoding of CD127 (IL-7-Rα; GTGTGAGGCACTCTT). The sorted single-cell suspensions were mixed and loaded

onto a Chromium single-cell device (10x Genomics) for encapsulation with barcoded gel beads according to the manufacturer's Single Cell 3' v3.1 (Dual Index) protocol. 3'GEX and 3' Feature barcoding libraries were prepared according to the standard manufacturer's protocol. The resulting libraries were sequenced on an Illumina NovaSeq 6000 S1 (paired end 150 bp).

## Bioinformatic analysis

The quality of sequencing data was assessed using FastQC (http://www.bioinformatics.babraham.ac.uk/projects/fastqc/). Reads were trimmed for adapters and low-quality base calls using Trim Galore, with default parameters (v0.6.5; https://www.bioinformatics.babraham.ac.uk/projects/trim_galore/). Reads were then mapped to the GRCm38 mouse reference genome using Hisat2 (v2.1.0[73];. BAM files were imported into Seqmonk (v1.47.0; http://www.bioinformatics.babraham.ac.uk/projects/seqmonk/), excluding those with mapping quality <30, and reads were quantified over merged mRNA isoforms from the GRCm38 v90 annotation, using the RNA-seq quantitation pipeline. Reads were also quantified specifically over the regions flanked by lox-P sites in *Zfp36l1* and *Zfp36l2*. Downstream analysis was performed in R (v4.1.2). Differential expression analysis comparing cKO with control samples was performed using DESeq2 (v1.22.2)[74], using 'normal' $\log_2$ fold change shrinkage, and genes designated differentially expressed if their FDR-adjusted p value was <0.05. Gene set enrichment analysis[75] was performed using the GSEAPreranked module (v7.4.0) of the GenePattern software package (v3.9_080823_b401)[76], with default parameters except that "collapse dataset" was set to "No_collapse". Genes were ranked based on log10 of their raw $P$ value from DESeq2 analysis, with a positive sign assigned to genes with a positive $\log_2$ fold change, such that the most significantly increased genes are ranked first, and the most significantly decreased are ranked last. Custom gene sets were uploaded for the analysis: the Hallmark apoptosis gene set was obtained from MSigDB (v6.2)[77], and gene names were converted to mouse orthologues using biomaRt[78] whilst the TCR signaling pathway gene set was curated manually (see Supplementary Data 4). Genes increased or decreased in Treg cells upon IL-2 or IL-7 treatment were identified using publicly available ImmGen common γ-chain cytokine RNA-seq data (GSE180020)[38], processed as above to compare cytokine- with PBS-treated Treg cells. Increased/decreased genes were defined as those with FDR-adjusted $P$ value < 0.05, and absolute $\log_2$-fold change >1, except for genes increased upon IL-7 treatment where a $\log_2$-fold change threshold of 0.7 was used to ensure a gene list of appropriate size for GSEA analysis. For IFNγ treatment, microarray data of iTreg cells treated for 10 h, compared with neutral conditions, was used (GSE38686[79]). Conditions were compared using GEO2R, and increased/decreased genes were defined as those with FDR-adjusted $P$ value < 0.05.

For heatmaps, normalized read counts were $\log_2$ transformed, and the mean transformed read count across all replicates for a given gene was subtracted; those with mean normalized read counts below 100 were excluded. Genes were ranked based on their adjusted $P$ value, and heatmaps were plotted using the pheatmap R package, with a threshold set on the fill color such that values above/below the maximum/minimum threshold were assigned the maximum/minimum colors in the scale.

To identify transcriptional regulators of endocytosis genes, ReMap (2022 release) DNA binding peaks[54] overlapping with 1 kb windows flanking the transcription start sites of all genes within the GSEA leading edge were downloaded from the UCSC genome browser; five expression-matched control gene sets were also generated, and peaks for these genes were downloaded in the same way. Transcriptional regulators were then filtered for those that were bound to at least 15 endocytosis genes or bound to at least 5 genes but additionally required this binding to be to more genes or represented by more peaks compared to all control gene sets. The identified transcriptional

regulators were ranked by FDR-adjusted *P* value from the DESeq2 comparison of *FYC⁺ l1l2* with *FYC⁺* nTreg cells, and the top 20 (excluding those with normalized read counts <100) were selected to display in the heatmap, as described above.

To generate lists of genes characteristic of naive and effector Treg cells (Supplementary Data 6), RNA-seq data from control nTreg and eTreg cells were compared as described above, and naive and effector markers were defined as genes with FDR-adjusted *P* value < 0.0001, and $\log_2$-fold change in eTreg cells compared with nTreg cells < −1.5 or >2, respectively.

R code underlying the analysis and figures has been deposited in GitHub (https://github.com/LouiseMatheson/Saenz-Narciso_Bell_2025; https://doi.org/10.5281/zenodo.15021467)[80].

## ScRNA-seq analysis

Since the feature barcoding libraries contained both multiplexing and CD127 tags, the fastq files were first split based on whether an exact match to one of the hashtag oligos for multiplexing was found starting in position 11 in the "R2" feature barcoding fastq file. The separated multiplexing and antibody-derived tag fastq files, together with the gene expression fastq files, were then processed using Cell Ranger v6.1.2, first with cellranger multi, using the GRCm38 mouse genome reference, followed by aggregation of the control and knockout data from the per sample outputs, using cellranger aggr. Further analysis was performed using the Seurat package (v4.1.0)[81] in R (v4.1.2). Cells containing below 1500 and over 14000 molecule counts for either hashtag 1 or hashtag 2 oligo-conjugated antibodies were removed to filter out putative empty droplets, or doublets, respectively. In addition, cells containing over 5.5% mitochondria-derived gene expression reads, over 10% reads originating from a single gene, or in which fewer than 1000 or more than 4200 genes were detected were removed. After filtering for quality, we compared the transcriptomes of 5824 cells from *FYC* mice and 4163 cells from *FYC l1l2* mice. Normalization was performed using the centered log ratio transformation, across cells (margin 2). The top 500 variable genes were identified, and these were scaled and used as input for principal component analysis. The top 15 principal components were then used as input for Seurat's graph-based clustering approach (FindNeighbors followed by FindClusters functions; resolution 0.5; all other parameters default). These 15 principal components were also used as input to RunUMAP for further dimensionality reduction and data visualization. To identify cluster-specific marker genes the FindMarkers function was used, only considering genes detected in at least 25% of cells in at least one group for a given comparison; significantly enriched or depleted genes for each cluster are listed in Supplementary Data 7. The SCINA package (v1.2.0) was used to assign cell type identities based on previous knowledge related to naive/effector cells (using the gene list in Supplementary Data 6). Genes associated with an *Ifng* signaling signature are listed in Supplementary Data 8; this was based on conversion of the human Reactome Interferon Gamma Signaling pathway genes to mouse orthologues. R code underlying the analysis and figures has been deposited in GitHub (https://github.com/LouiseMatheson/Saenz-Narciso_Bell_2025; https://doi.org/10.5281/zenodo.15021467)[80].

## Identification of direct ZFP36-family targets

HITS-CLIP data for ZFP36-family proteins in CD4⁺ T cells following 4 h activation, or 72 h activation with 2 h reactivation, were obtained from GSE96074[9] iCLIP data for ZFP36L1 in CD4⁺ T cells activated for 24 h with anti-CD3 and anti-CD28 was obtained from GSE155087[10]. All data was analyzed using the iCount pipeline (v2.0.1.dev)[82] on the Genialis platform (now obsolete: iCount now hosted on https://app.flow.bio); for iCLIP data the replicates for each antibody were merged. A gene was designated a target if the 3′UTR contained a significant crosslink site (FDR < 0.05) with both antibodies in the iCLIP, or if a crosslink site in the 3′UTR was identified in at least two replicates for either of the

HITS-CLIP datasets. CLIP crosslinks over the 3′UTRs of selected transcripts were visualized using a shiny app (https://github.com/LouiseMatheson/iCLIP_visualisation_shiny; https://doi.org/10.5281/zenodo.14982514)[83], without filtering for significant sites.

## Statistical analysis

Statistical significance was determined using GraphPad Prism v9 using the test indicated in the respective Figure legends. All Mann–Whitney tests performed were two-tailed.

## Reporting summary

Further information on research design is available in the Nature Portfolio Reporting Summary linked to this article.

## Data availability

The RNA-seq data generated in this study have been deposited in the Gene Expression Omnibus (GEO) database under accession code GSE244621. The cytokine RNA-seq and microarray data used in this study are available in the GEO database under accessions GSE180020 and GSE38686. The CLIP data used in this study are available in the GEO database under accessions GSE96074 and GSE155087. Mice with modified alleles of *Zfp36*, *Zfp36l1*, and *Zfp36l2* are available under a material transfer agreement with The Babraham Institute. All processed data needed to evaluate the conclusions are presented in the paper or in the Supplementary Materials/Source Data file. Source data are provided with this paper.

## Code availability

The R code underlying the RNA-seq and iCLIP analysis and visualization has been deposited in GitHub, and released with Zenodo under the https://doi.org/10.5281/zenodo.15021467 [https://doi.org/10.5281/zenodo.15021467][80] and https://doi.org/10.5281/zenodo.14982514 [https://doi.org/10.5281/zenodo.14982514][83].

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

## Acknowledgements

The authors thank Kirsty Bates for expert technical assistance and Oliver Burton for advice on flow cytometry; Fiamma Salerno and Marian Jones Evans for characterizing the reporter mice; and the Core Biochemical Assay Laboratory at Addenbrooke's Hospital for MSD analysis. The authors thank Georg Petkau and Arianne Richard for critical reading of the manuscript. The authors thank the UKRI-BBSRC Core Capability Grant funded Babraham Institute Biological Support Unit, Sequencing, Flow Cytometry and Bioinformatics Facilities for invaluable support. This study was funded by the BBSRC Institute strategic programme grants BBS/E/B/000C0427; BBS/E/B/000C0428 and a Wellcome Investigator award (200823/Z/16/Z) to M.T.

## Author contributions

Conceptualization: M.T. Methodology: B.S.-N. and S.E.B. Investigation: B.S.-N., S.E.B., and R.V. Data curation: B.S.-N. and L.S.M. Writing—original draft: B.S.-N., S.E.B., L.S.M., and M.T. Funding acquisition: M.T. Supervision: M.T. All authors contributed to reviewing and editing the manuscript.

## Competing interests

M.T. has a funded collaboration with AZ on a topic unrelated to this study. The remaining authors declare no competing interests.
