## [Transparent Peer Review file · Nature Communications]

ZFP36-family RNA-binding proteins in regulatory T cells reinforce immune homeostasis.

Corresponding Author: Dr Martin Turner

Version 0:

Reviewer comments:

Reviewer #1

(Remarks to the Author)

Review of MS: NCOMMS-24-61322-T entitled "ZFP36-family RNA-binding proteins in regulatory T cells reinforce immune homeostasis". The study by Sáenz-Narciso and colleagues investigates combined deletion of Zfp36l1 and Zfp36l2 in Treg cells. Different from deletion with CD4-Cre, deletion with Foxp3-IRES-YFP-Cre causes the induction of effector memory CD4 and CD8 T cells and the development of fatal autoimmune or autoinflammatory disease. The disease manifests for example in a colitis phenotyp. The authors set out to find out how Treg cells are changed in their gene expression and function, and they make a number of remarkable observations including differential CTLA-4 localization, hyposensitivity to IL-2 and IL-7 and hypersensitivity to lfn-g signaling. Finally, the authors rescue the fatal disease by genetic combination with one lfn-g knockout allele. The manuscript shows a thorough investigation and clear presentation, although the direct relationship which targets cause which phenotypes remain unsolved, however, providing such causal proof may exceed the scope of this paper. I only have a few points to help improve the paper.

Major points

1. The authors make elegant approaches to compare what happens in Treg cells when Zfp36l1 and Zfp36l2 are knocked out– either in all Tregs of (male) mice, or in a fraction of Treg cells – in (female) mice. This approach should also be used to ask which Treg intrinsic phenotypes are rescued by heterozygous lfn-g deletion. Is the competitive fitness restored, how are the phenotypes of Tfr, eTreg affected.?
2. The authors intersect differentially regulated genes in Treg cells with the identified targets of Zfp36l1/l2 proteins in CD4+ and CD8+ T cells, and they identify interesting categories. Although this is done in a very sophisticated manner, one does not get to know the drivers of the phenotypes. Within the deregulated targets, one would expect the stronger deregulated targets having stronger influences on the observed phenotype. Nevertheless, the authors seem to argue that numerous subtle changes together build the very pronounced phenotype. It would be informative if targets that can explain the phenotype, were also tested for deregulation in the right cell type/activation stage etc.. For example, are the SOCS family members involved in cytokine sensitivity? Which genes/targets can be involved in CTLA-4 localization?
3. The authors prevent the development of disease with inactivation of one lfn-g allele, a prototypic Zfp36 target with functional AU-rich elements. However, this approach does not easily explain why inactivation of Zfp36l1 and Zfp36l2 with CD4-Cre does not cause a similar disease, if the excess of lfn-g originates from conventional T cells. One possible explanation for the observed phenotype could be that ex-Treg cells with autoreactive TCRs develop, if Zfp36l1/l2 RBPs are inactivated with Foxp3-Cre. Those may be superior, tissue localized lfn-g producers, since lfn-g expression is induced in the exTregs by TCR signals, and is no longer repressed by Zfp36 proteins binding to AU-rich elements. Ex-Tregs would have downregulated YFP and would therefore not be captured in the approaches shown in the paper so far. This possibility should be addressed experimentally, or at least mentioned in the "Limitations of this study" section.

Reviewer #2

(Remarks to the Author)

The study „ZFP36-family RNA-binding proteins in regulatory T cells reinforce immune homeostasis” by Sáenz-Narciso and Bell et al. investigates the roles of Zfp36, Zfp36l1 and Zfp36l2 in mouse Treg cells. The authors use conditional Foxp3-specific knock-out mice for each factor (or double-KO for Zfp36l1 and Zfp36l2) to determine their individual roles. While deletion of Zfp36 or Zfp36l2 had no apparent effect, the absence of Zfp36l1 in Tregs caused a systemic hyper immune activation, which was even more pronounced when Zfp36l1 and Zfp36l2 were both deleted at the same time.

Mechanistically, the authors use bulk RNA-Seq and scRNA-Seq of Zfp3611/12-deficient Tregs to reveal potential underlying mechanisms. While the core Treg signature appears stable in KO Treg cells, decreased expression of CD25 and CD127 seems to dampen IL-2/IL-7 sensitivity of KO Tregs, likely dampening their fitness. In contrast, KO Tregs respond more to IFN γ , while also producing more IFN γ themselves. Finally, the authors show that most of the hyper immune phenotype of Treg-specific Zfp3611/12-deficient mice is due to excessive IFN γ signalling, since genetic deletion of one allele of Ifng is sufficient to restore homeostasis.

In summary, the study is likely to be of considerable interest for the readers in the field. However, the authors missed to work out the precise molecular mechanisms and targets of how Zfp36 family members control Treg cell function. Also it remains mechanistically unclear, why Zfp3611/12-deficient Tregs fail to control especially type 1 IFN γ -producing T cells. Therefore, the paper mostly provides a comprehensive characterization of the respective KO mice.

Specific points:

Figure 1:

The authors report that 9% of FYCI1 mice and 12% of FYCI1/2 develop symptoms. And as far as I understand, the authors focus their subsequent analysis entirely on these sick mice. What about the majority of mice that don't develop symptoms? Are they completely healthy? Do they show any differences in effector CD4⁺/CD8⁺ cells? How do the Treg cells look like in these mice?

It would be helpful to show a more comprehensive analysis of key clinical parameters (e.g. weight change over time, pathology scores) for all mice and (immune) phenotyping for symptomatic vs. non-symptomatic mice.

Overall, the unanswered question remains: Why do some KO mice experience a loss of immune homeostasis, while others not?

Figure 2:

I am missing a clearly unbiased analysis of the bulk RNA-Seq data from WT vs. KO Tregs. Right now, the authors only use a custom pre-defined group of gene sets for their enrichment analysis, to basically pave the way for their findings regarding IL-2/IL-7 sensitivity. It feels like these findings were already known before the RNA-Seq.

I think a more explorative approach using unbiased GSEA for example could reveal additional signaling pathways or metabolic alterations in KO Tregs, that might contribute to the diminished fitness and function of Zfp3611/12-deficient Tregs. Additionally, it remains unclear which molecular targets are specifically bound by Zfp3611/12 in Treg cells. The authors use published CLIP data from activated CD4⁺ T cells, however CLIP data directly from Treg cells would certainly be more helpful to elucidate the molecular mechanism. Such data would also be a valuable resource for the community.

Figure 7:

The analysis of cytokine expression by Tcon and Treg in Fig. 7i is really informative and reveals that there is a major dysregulation of type 1 IFN- γ -producing effector T cells in FYCI 1/2 mice, while type 2 and type 3 immunity seems rather unaffected. It also shows that KO Tregs do not have a problem in producing IL-10, ruling out this as a potential mechanism underlying the breakdown of immune homeostasis.

There is the concept that type 1 T cells (eg Th1) are primarily regulated by Th1-like Treg cells, that mimic Th1 cells in terms of e.g. T-bet expression etc. The question that comes to my mind here is: Do KO Tregs cannot acquire a proper and functional Th1-like Treg phenotype and is this the reason for their reduced capacity to suppress type 1 immune cells?

I am asking this bc the paper is overall lacking the explanation why Zfp3611/12-deficient Tregs cannot properly suppress (Can they actually suppress normally in conventional in vitro suppression assays?) and why specifically not type 1 immune cells. More data addressing this question would certainly increase the scope of the paper immensely.

Minor comment on Figure 7i and Sup. Fig. 8. The staining of ROR γ t⁺ Tregs doesn't really show any pos. population to me. I suggest to stain for these Treg subsets (also the GATA3⁺ Treg) in intestinal Treg cells rather than Treg cells from Spleen/LN.

Reviewer #3

(Remarks to the Author)

In this study, Saenz-Narciso and colleagues investigate the role of ZFP36 family RBPs in Tregs. They specifically ablate Zfp3611, Zfp3612, and Zfp3611/2 in Tregs and find that the loss of either protein or the combined loss of both leads to autoimmunity, as evidenced by the expansion of activated CD4 and CD8 T cells expressing proinflammatory cytokines. Mechanistically, these RBPs regulated multiple aspects of Treg biology, including endocytosis and sensitivity to cytokines, both of which had significant implications for Treg stability and function. The authors further demonstrated that IFN γ drives the inflammatory phenotype in mice lacking Zfp3611/2 specifically in Tregs, noting that deletion of one allele of IFN γ is sufficient to ameliorate this phenotype. This is an excellent study with high-quality data; however, a few issues require clarification.

1. The authors state that only 9% of FYCI1 and 13% of FYCI1/2 mice develop autoimmunity. Why is the disease penetrance not 100%? It appears the authors focus solely on male mice. Is there a specific reason for not analysing homozygous knockout females?

2. In female FYCI1I2/+ mice, the Foxp3+YFP+ population was outcompeted by Foxp3+YFP-WT Tregs, suggesting that Tregs lacking I1I2 have reduced competitive fitness. Do knockout Tregs exhibit an activated phenotype in this context (with competition), or is there a skewed nTreg/eTreg ratio?

3. At the RNA level, there is downregulation of Ctla4, which appears to correlate with low protein levels of total CTLA4, despite the claim of endocytosis. What is the underlying cause of the downregulation of CTLA4 protein?

4. Is defective STAT5 signalling responsible for the lack of competitive fitness in FYCI1I2 mice? Can the IL-2/Ab complex expand YFP+ Tregs in FYCI1I2/+ mice?

5. FYCI1I2 Tregs produce more IFN γ and IL-10 when stimulated. Is this phenotype a result of inflammation in these mice? Would YFP+ Tregs in FYCI1I2/+ mice display the same phenotype? A major concern is distinguishing the actual phenotype of Tregs caused by I1I2 deletion from that induced by inflammation. For example, bulk RNA sequencing has been performed on Tregs from FYCI1I2/+ mice (with no inflammation), while scRNA sequencing was conducted on Tregs from FYCI1I2 mice (where inflammation is high). Can these two datasets be compared (using scRNA-seq pseudobulk analysis) to differentiate between inflammation-induced and genuine I1I2-dependent gene expression?

6. What happens to Tregs in non-lymphoid tissues? Are there any defects in migration?

Version 1:

Reviewer comments:

Reviewer #1

(Remarks to the Author)

The authors have adequately addressed all of my points and provided important new information.

Reviewer #2

(Remarks to the Author)

The authors have addressed all my concerns. Thank you.

Regarding Figure 1, I would suggest to include the histology data (now Sup. Fig. 1c) and the LN cellularity data (now Sup. Fig. 1d and 2a) into the main Figure, bc they are strong phenotypic data that don't need to be hidden in the supplement.

The potential reasons why only a minor fraction of mice develops clinically symptoms, should be briefly discussed in the manuscript.

Reviewer #3

(Remarks to the Author)

The authors have addressed all of my queries. I have no further concerns. This is an excellent study!

Reviewer #1 (Remarks to the Author)

Review of MS: NCOMMS-24-61322-T entitled “ZFP36-family RNA-binding proteins in regulatory T cells reinforce immune homeostasis”. The study by Sáenz-Narciso and colleagues investigates combined deletion of *Zfp36l1* and *Zfp36l2* in Treg cells. Different from deletion with CD4-Cre, deletion with *Foxp3*-IRES-YFP-Cre causes the induction of effector memory CD4 and CD8 T cells and the development of fatal autoimmune or autoinflammatory disease. The disease manifests for example in a colitis phenotype. The authors set out to find out how Treg cells are changed in their gene expression and function, and they make a number of remarkable observations including differential CTLA-4 localization, hyposensitivity to IL-2 and IL-7 and hypersensitivity to *Ifn-g* signaling. Finally, the authors rescue the fatal disease by genetic combination with one *Ifng* knockout allele. The manuscript shows a thorough investigation and clear presentation, although the direct relationship which targets cause which phenotypes remain unsolved, however, providing such causal proof may exceed the scope of this paper. I only have a few points to help improve the paper.

We thank the reviewer for their positive comments and feedback on our manuscript which we have taken on board to improve the study and its presentation, we respond to each point below. Revisions/new text are highlighted in the revised manuscript.

Major points

1. The authors make elegant approaches to compare what happens in Treg cells when *Zfp36l1* and *Zfp36l2* are knocked out– either in all Tregs of (male) mice, or in a fraction of Treg cells – in (female) mice. This approach should also be used to ask which Treg intrinsic phenotypes are rescued by heterozygous *Ifng* deletion. Is the competitive fitness restored, how are the phenotypes of *Tfr*, *eTreg* affected?

In the submitted version of the manuscript, we did not present data on the *eTreg* populations in *FYC⁺ I112* female mice. In these mice, the proportions of *eTregs* amongst *YFP⁺ Tregs* were not different than those in *FYC⁺* mice; thus, the expansion of *eTreg* in *FYC I112* male mice is not a Treg intrinsic phenotype. We have now added this data as **Supplementary fig. 3a**.

We have assessed whether the competitive disadvantage of *YFP⁺ Tregs* observed in *FYC⁺ I112* mice is affected by heterozygous *Ifng* deletion. In *Ifng^{+/-} FYC⁺ I112* mice the proportions and numbers of *YFP⁺ Tregs* are not different to those in *FYC⁺ I112* mice indicating that loss of one allele of *Ifng* does not restore the competitive fitness (**Reviewer Figure 1**).

Reviewer Figure 1. Loss of one allele of *Ifng* does not rescue the competitive disadvantage observed in *YFP⁺ Tregs* from *FYC⁺ I112* mice. Flow cytometry plots showing *FOXP3* versus *YFP* expression in *FYC⁺*, *FYC⁺ I112* and *Ifng^{+/-} FYC⁺ I112* mice (left panel); proportions and numbers of Tregs gated as *CD4⁺ FOXP3⁺ YFP⁻* or *CD4⁺ FOXP3⁺ YFP⁺* (right panel); key as shown.

Additionally, we observed no difference in *CD25* or *CD127* expression between *YFP⁺ Tregs* from *Ifng^{+/-} FYC⁺ I112* and *FYC⁺ I112* mice (**Reviewer Figure 2**). Thus, deletion of one allele of *Ifng* is insufficient to restore expression of these cytokine receptors.

Reviewer Figure 2. Loss of one allele of *Ifng* does not rescue CD25 nor CD127 expression levels in YFP⁺ Tregs from *FYC*^{+/+} *I112* mice. a) Representative histogram overlays of CD25 expression and CD25 gMFI on nTreg (left panel) and eTreg (right panel) from spleen; b) Representative histogram overlays of CD127 expression and CD127 gMFI on nTreg (left panel) and eTreg (right panel) from spleen; n=3; key as shown.

We propose not to include these results in the revised manuscript.

2. The authors intersect differentially regulated genes in Treg cells with the identified targets of *Zfp3611/2* proteins in CD4⁺ and CD8⁺ T cells, and they identify interesting categories. Although this is done in a very sophisticated manner, one does not get to know the drivers of the phenotypes. Within the deregulated targets, one would expect the stronger deregulated targets having stronger influences on the observed phenotype. Nevertheless, the authors seem to argue that numerous subtle changes together build the very pronounced phenotype. It would be informative if targets that can explain the phenotype, were also tested for deregulation in the right cell type/activation stage etc.. For example, are the SOCS family members involved in cytokine sensitivity? Which genes/targets can be involved in CTLA-4 localization?

ZFP36 proteins can regulate mRNA at multiple levels by direct binding to AU-rich elements in the 3'UTR of mRNAs. Thus, it is possible that multiple mechanisms contribute to the complex phenotype and only the effects on RNA decay can be inferred from our RNAseq data.

Given the defect in CTLA-4 cycling we have performed a deeper analysis of the genes within the endocytosis gene set that are differentially expressed between naïve Treg from *FYC*^{+/+} and *FYC*^{+/+} *I112* mice and that are present in the leading edge of our GSEA analysis (i.e., genes that show a trend towards increased expression). These data are now included in the new **Supplementary Fig. S10 and S11** and we have added additional text to the results.

In addition, we found that several genes that might play a role in endocytosis, and/or CTLA-4 cycling are not directly targeted by the ZFP36 family which prompted us to examine transcription factors and epigenetic regulators that might be important in their regulation. We identified transcriptional regulators that were frequently bound at the promoters of genes within the endocytosis GSEA leading edge (**Supplementary Fig. 12**). Many transcription factors were ZFP36-family targets by CLIP and several were known to have roles in Tregs, including *Ets1*, *Stat1*, *Hopx*, and *Bcl11b*.

Overall, we identified numerous genes whose expression is altered by deletion of *Zfp361/1/2* that have the potential to impact on endocytosis, including both direct targets of the ZFP36 family, and genes regulated indirectly. How this complex network of alterations coalesces to result in the decreased cycling of CTLA-4 in *Zfp361/Zfp362* KO Tregs will be an important area for future in-depth study.

Regarding the changes in cytokine sensitivity, no SOCS family member showed a relevant increase at the mRNA level (**Reviewer Figure 3**).

Reviewer Figure 3. RNAseq from the SOCS family.

Normalised read counts of the SOCS family members in YFP⁺ nTreg from *FYC⁺* and *FYC⁺I1I2* mice; n=6-7; key as shown. Read counts were normalised using size factors derived from the overall DESeq2 analysis of all genes; p values determined using t-test with FDR correction.

We observed only a minor decrease (1.3-fold change) in the expression of SOCS1 protein by flow cytometry in eTregs from *FYC I1I2* male mice (**Reviewer Figure 4a**). In our opinion, this is unlikely to account for the diminished signalling in Treg in the knockout mice in response to IL-2 and IL-7 but may contribute to the increased sensitivity to IFN γ (<https://pmc.ncbi.nlm.nih.gov/articles/PMC11700635/>). To further investigate if the diminished SOCS1 levels were cell intrinsic we analysed LN from *FYC⁺ I1I2* female mice. We observed no difference in SOCS1 levels between YFP⁺ control Tregs and YFP⁺ Tregs lacking *I1I2* (**Reviewer Figure 4b**). Thus, the diminished SOCS1 levels observed in *FYC I1I2* male mice are likely due to the inflammatory environment.

Reviewer Figure 4. SOCS1 expression is not affected by the lack of Zfp3611 and Zfp3612.

- a) Representative histogram overlays of SOCS1 expression in nTreg (left panel) and eTreg (right panel) from LN from male *FYC I1I2* and *FYC* mice; gMFI of SOCS1.
- b) Representative histogram overlays of SOCS1 expression in YFP⁺ nTreg (left panel) and eTreg (right panel) from LN from female *FYC^{+/+}* and *FYC^{+/+} I1I2* mice; gMFI of SOCS1; key as shown.

Further in-depth analysis of the SOCS family is limited by the lack of reagents for detecting these proteins with the required sensitivity and we suggest additional experiments to investigate this area are beyond the scope of this manuscript.

3. The authors prevent the development of disease with inactivation of one *lfn* allele, a prototypic *Zfp36* target with functional AU-rich elements. However, this approach does not easily explain why inactivation of *Zfp3611* and *Zfp3612* with *CD4-Cre* does not cause a similar disease, if the excess of *lfn* originates from conventional T cells. One possible explanation for the observed phenotype could be that ex-Treg cells with autoreactive TCRs develop, if *Zfp3611/2* RBPs are inactivated with *Foxp3-Cre*. Those may be superior, tissue localized *lfn* producers, since *lfn* expression is induced in the exTregs by TCR signals, and is no longer repressed by *Zfp36* proteins binding to AU-rich elements. Ex-Tregs would have downregulated YFP and would therefore not be captured in the approaches shown in the paper so far. This possibility should be addressed experimentally, or at least mentioned in the "Limitations of this study" section.

One key difference between the *CD4cre* system and the *Foxp3cre* system we are using in our study is that in the *CD4cre* system *Zfp3611* and *Zfp3612* are absent in naïve and effector CD4 and CD8 T cells as well as Treg. In the manuscript by Cook et al. (<https://pmc.ncbi.nlm.nih.gov/articles/PMC9832469/>) their Figure 8 shows that in in *Cd4-Cre⁺ Zfp3611^{fl/fl} Zfp3612^{fl/fl}* mice there is defective priming of CD4 T cells. The mutant T cells are unable to generate an immune response and Cook et al. show *Cd4-Cre⁺ Zfp3611^{fl/fl} Zfp3612^{fl/fl}* mice are resistant to the induction of experimental autoimmune encephalomyelitis (EAE).

We have added new data (**Figure 9**) which addresses the reviewer's question of whether Tregs that lack *Zfp36l1* and *Zfp36l2* can give rise to ex-Tregs.

In *Foxp3^{eGFP-creERT2} R26^{flSTOP-tdRFP} 11/2* mice we observed an increase in the number of ex-Tregs, indicating that these RBP are required for the maintenance of Treg stability (**Figure 9a**). The ratio of ex-Treg/Treg was higher in female *Foxp3^{eGFP-creERT2}* heterozygous *R26^{flSTOP-tdRFP} 11/2* mice (**Figure 9h**), suggesting that the instability observed in *11/2* deficient Tregs is cell intrinsic.

Reviewer #2 (Remarks to the Author)

The study „ZPF36-family RNA-binding proteins in regulatory T cells reinforce immune homeostasis” by Sáenz-Narciso and Bell et al. investigates the roles of *Zfp36*, *Zfp36l1* and *Zfp36l2* in mouse Treg cells. The authors use conditional *Foxp3*-specific knock-out mice for each factor (or double-KO for *Zfp36l1* and *Zfp36l2*) to determine their individual roles. While deletion of *Zfp36* or *Zfp36l2* had no apparent effect, the absence of *Zfp36l1* in Tregs caused a systemic hyper immune activation, which was even more pronounced when *Zfp36l1* and *Zfp36l2* were both deleted at the same time.

Mechanistically, the authors use bulk RNA-Seq and scRNA-Seq of *Zfp36l1/2*-deficient Tregs to reveal potential underlying mechanisms. While the core Treg signature appears stable in KO Treg cells, decreased expression of CD25 and CD127 seems to dampen IL-2/IL-7 sensitivity of KO Tregs, likely dampening their fitness. In contrast, KO Tregs respond more to IFN γ , while also producing more IFN γ themselves. Finally, the authors show that most of the hyper immune phenotype of Treg-specific *Zfp36l1/2*-deficient mice is due to excessive IFN γ signalling, since genetic deletion of one allele of *Ifng* is sufficient to restore homeostasis.

In summary, the study is likely to be of considerable interest for the readers in the field. *However, the authors missed to work out the precise molecular mechanisms and targets of how Zfp36 family members control Treg cell function.* Also, it remains mechanistically unclear why *Zfp36l1/2*-deficient Tregs fail to control especially type 1 IFN γ -producing T cells. Therefore, the paper mostly provides a comprehensive characterization of the respective KO mice.

We thank the reviewer for their positive comments and feedback on our manuscript which we have taken on board to improve the study and its presentation, we respond to each point below. Revisions/new text are highlighted in the revised manuscript.

Specific points:

Figure 1:

The authors report that 9% of *FYCI1* mice and 12% of *FYCI1/2* develop symptoms. And as far as I understand, the authors focus their subsequent analysis entirely on these sick mice. **What about the majority of mice that don't develop symptoms? Are they completely healthy? Do they show any differences in effector CD4⁺/CD8⁺ cells? How do the Treg cells look like in these mice?**

It would be helpful to show a more comprehensive analysis of key clinical parameters (e.g. weight change over time, pathology scores) for all mice and (immune) phenotyping for symptomatic vs. non-symptomatic mice.

Overall, the unanswered question remains: Why do some KO mice experience a loss of immune homeostasis, while others not?

We have now clarified in the text the health status of the animals analysed. The majority of the data presented in the original version of the manuscript, except the histology shown in **Figure 1e** and the Kaplan Meier plot shown in **Figure 8a**, had been obtained from apparently healthy mice. For clarity, we have edited the text and moved the histology data from mice with clinical symptoms to **Supplementary Fig. 1c**. We now include data showing lymph node cellularity is increased in

mice with clinical signs (**Supplementary Fig. 1d**). All of the apparently healthy male *FYC 11/2* mice analysed also showed increased lymph node cellularity and increased proportions and numbers of CD4 and CD8 effector cells, GC B cells, and eTregs. Thus, all male mice have shown evidence for the loss of immune homeostasis but only some of them progressed to exceed the clinical severity limit by the time we analysed them.

We can only speculate as to why 9-12% mice develop symptoms that exceed the severity limit, but consider potential effects from the local cage environment and the aggressive interactions between male mice that are cohoused, together with the overall specific-pathogen-free facility in which they are housed. The gut flora was altered Schaedler flora and since the opening of this barrier facility in 2009, no primary pathogens or additional agents listed in the FELASA recommendations (<https://pubmed.ncbi.nlm.nih.gov/24496575/>) have been confirmed during health monitoring surveys of the stock holding rooms. The transition from mild to moderate severity is governed by a number of assessment criteria as listed in the attached PDF (Limiting clinical signs in laboratory rodents), with marked piloerection and intermittent hunching being the most common limiting adverse effect resulting in humane killing of an animal. We have attached this as a Supplemental file to the resubmission.

Figure 2: *I am missing a clearly unbiased analysis of the bulk RNA-Seq data from WT vs. KO Tregs. Right now, the authors only use a custom pre-defined group of gene sets for their enrichment analysis, to basically pave the way for their findings regarding IL-2/IL-7 sensitivity. It feels like these findings were already known before the RNA-Seq. I think a more explorative approach using unbiased GSEA for example could reveal additional signaling pathways or metabolic alterations in KO Tregs, that might contribute to the diminished fitness and function of Zfp361/12-deficient Tregs.*

We believe that the reviewer's definition of a 'clearly unbiased' analysis would comprise, for example, a GSEA analysis using a curated group of gene sets such as KEGG, Hallmark, Reactome or Gene Ontology. During the exploration of our RNA-seq data, we performed numerous such analyses to aid us in identifying cellular functions and pathways that were altered in *Zfp361/12*-deficient Tregs and warranted further investigation. We considered showing one of these in our manuscript, however, we decided to show a more focussed group of gene sets with direct relevance to T cells. We believe that this still represents a global analysis; whilst we may in part have been guided by the larger-scale analyses, we aimed to include gene sets for the major signalling and metabolic pathways, as well as numerous aspects of cell function, including DNA and RNA metabolism, the cell cycle and apoptosis, and gene sets more specific to T cell differentiation and function. The breadth of the gene sets chosen can be seen in **Supplementary Table 4**. Our decision to use this more focussed group of gene sets was based on a number of considerations:

1. In our view, all gene set analyses are, by their nature, biased. For any curated group of gene sets, decisions will have been made on what those sets comprise, and the genes within them will be biased, for example, towards more widely studied cell types. We would, therefore, argue that the advantage of such curated gene sets in being unbiased is not as clear-cut as implied by the reviewer.
2. No curated group of gene sets covers everything that might be of interest. Hallmark gene sets are the most limited; however, even for the larger groups of gene sets, this is an issue. For example, KEGG does not contain a general "Apoptosis" gene set, whilst Reactome does not include a general "cell adhesion" or "migration" gene set. None of these groups of gene sets include a "Treg signature".
3. The interpretation of large-scale gene set analyses is hampered by the inclusion of many irrelevant gene sets, as well as by the huge amount of redundancy inherent to gene ontology terms, and to a lesser extent to Reactome and KEGG pathways; together this means that the reader has to dig into the genes within each set to understand what they

mean, and plough through a lot of repeated information to reach additional interesting pathways that are buried beneath. The redundancy can, in part, be mitigated by tools such as REVIGO for gene ontology terms, but in our experience, substantial redundancy often still remains. We show, as an example (**Reviewer Figure 5**), the top 15 increased and decreased gene sets from an analysis of all KEGG pathways. The increased pathways include “platelet activation”, “cardiomyopathy”, “axon guidance” and “collecting acid duct secretion”, none of which are directly relevant to Tregs. “Human cytomegalovirus infection” and “Human papilloma virus infection” likely relate to altered immune cell function – but in our context are clearly not indicating that the cells are infected with these viruses. Regulation of the actin cytoskeleton is driven by increased expression of integrin genes, likely reflecting altered signalling and/or cell migration, both of which are covered more specifically by the pathways that we focus on. The decreased pathways show some redundancy, with two gene sets related to ribosome biogenesis, two to DNA repair, and several to amino acid metabolism. All of these examples require substantial additional effort to understand and interpret, and/or push other gene sets of interest out of the top few that are possible to show in a figure. Whilst such effort in interpretation was necessary for us in exploring and understanding our data, for a manuscript figure we preferred to show a more focussed analysis, enabling much clearer interpretation by the reader.

Reviewer Figure 5. GSEA from KEGG pathways

Top 15 increased and decreased pathways from all the KEGG pathways. Numbers in brackets indicate the total number of genes in the pathway; numbers in white indicate FDR-adjusted p values.

- Except for Hallmark gene sets, which have a relatively limited scope, none of the curated groups of gene sets are designed for GSEA, and thus, they include many gene sets that are either too large (in the case of gene ontology) or too small. We note that, in the KEGG example shown, many of the decreased pathways are close to the lower limit of 15 genes acceptable for GSEA, and that even above this limit, enrichment of such gene sets can be driven by very few genes and is more subject to artefacts. In some cases, our desire to include a particular pathway meant that a small gene set was unavoidable, with the smallest gene set we included comprising 23 genes (One-carbon metabolism). However, we aimed to minimise this and included only two gene sets with less than 30 genes, with the vast majority of our gene sets comprising between 50-250 genes.

5. Performing the analysis in this way allowed us to add in gene sets that were of particular interest to us, such as the Treg signature, and an improved T cell signalling pathway gene set which we have curated based on our knowledge of T cell signalling. As the reviewer suspects, we were aware of the effects on cytokine signalling when we assembled our group of gene sets, and we included the gene sets derived from immunological genome project data for IL-2 and IL-7, and published data for IFN γ , in order to test the hypothesis that altered signalling impacts on downstream gene expression. This enabled us to use data derived from Tregs, and thus directly relevant to the signalling responses in our cells. The *highly significant results* we obtained using this data would not have been seen from any of the curated lists of gene sets, which either contain gene sets comprising the signalling machinery (KEGG and Reactome), or include very generic gene sets for altered or increased gene expression upon IL-2/STAT-5 or IFN γ signalling (Hallmark and GO biological process). We considered carefully whether we should include these gene sets together with the global GSEA, or in a separate analysis, alongside our other data on cytokine signalling. We chose to include them here because we wanted to be able to compare the GSEA statistics of these sets to the other gene sets included. This would not be possible if the analyses were performed separately, since, whilst the values may be broadly similar, the way in which the normalised enrichment scores and p-values are calculated means that they would not be directly comparable.

We acknowledge that there will be additional altered pathways that do not appear in the analysis we show. However, we believe that the differences between the available curated sets, and the issues outlined above mean that no large-scale GSEA analysis is able to both comprehensively and concisely reveal all pathways with alterations. Our chosen group of gene sets provides good coverage of relevant pathways, whilst mitigating some of the problems inherent to such analyses.

Additionally, it remains unclear which molecular targets are specifically bound by Zfp361/12 in Treg cells. The authors use published CLIP data from activated CD4+ T cells, however CLIP data directly from Treg cells would certainly be more helpful to elucidate the molecular mechanism. Such data would also be a valuable resource for the community.

We agree that iCLIP from Tregs would be perfect for identifying ZFP36L1/L2 targets. However, unfortunately the small numbers of Tregs that can be obtained *ex vivo* preclude the generation of good quality iCLIP data. Given this limitation, we believe that CLIP data from activated CD4+ T cells provides a good surrogate and will allow us to identify the majority of direct targets. Use of CLIP data from three different activation conditions also aimed to maximise our detection of target transcripts. Comparison of our target list generated from CD4+ T cells with a similar list generated using CD8+ cytotoxic T lymphocyte (CTL) iCLIP (<https://doi.org/10.1038/s41467-022-29979-x>) shows a large degree of overlap between these (**Reviewer Figure 6a**). Moreover, comparison of this dataset with the 4h-activated CD4+ CLIP data shows that there is a very high correlation between the sum of crosslinks over gene 3'UTRs between CD4+ and CD8+ T cells, with genes significant in only one of the datasets typically showing weaker detection (**Reviewer Figure 6b**). Whilst the correlation will, in part, be driven by the correlation between expression levels, this is itself a major determinant of whether a target is detected in a CLIP experiment. We would expect Treg CLIP data to show a similar, if not greater, correspondence with activated CD4+ T cell CLIP data.

Reviewer Figure 6 CD4 and CD8 CLIP targets are correlated.

a) Venn Diagram showing the overlap between CD4 and CD8 (CTL) CLIP targets

b) Plot showing the correlation between the sum of crosslinks over gene 3'UTRs between CD4⁺ (4h-activated dataset) and CD8⁺ (CTL) T cells

We do acknowledge that it is possible that some direct targets of ZFP36L1/L2 will be missed in the activated CD4⁺ T cell data, either due to Treg-specific gene expression, or differential targeting of a transcript in Treg compared to activated CD4⁺ T cells; this may also result in genes detected in the activated CD4⁺ T cells that are not targeted in Tregs. Therefore, in order to strengthen our analysis, we have identified transcripts that have an AU-rich element (defined as 2 x UAUU separated by up to 3 nucleotides) within their 3'UTR, providing a potential high-affinity binding site for the ZFP36 family. We have revised **Fig. 2c,d,e**, **Supplementary Fig. 4**, **Supplementary Fig. 5a** and **Supplementary Tables 4** and **5**, to incorporate this information. Whilst the presence of a potential AU-rich element does not necessarily mean that a transcript is a target, we believe that our approach is helpful for identifying genes that may be direct targets, as well as providing additional evidence for genes identified in the CLIP data.

Figure 7: The analysis of cytokine expression by Tcon and Treg in Fig. 7I is really informative and reveals that there is a major dysregulation of type 1 IFN-g-producing effector T cells in FYC1/2 mice, while type 2 and type 3 immunity seems rather unaffected. It also shows that KO Tregs do not have a problem in producing IL-10, ruling out this as a potential mechanism underlying the breakdown of immune homeostasis.

There is the concept that type 1 T cells (eg Th1) are primarily regulated by Th1-like Treg cells, that mimic Th1 cells in terms of e.g. T-bet expression etc. The question that comes to my mind here is: Do KO Tregs cannot acquire a proper and functional Th1-like Treg phenotype and is this the reason for their reduced capacity to suppress type 1 immune cells?

I am asking this bc the paper is overall lacking the explanation why Zfp36l1/l2-deficient Tregs cannot properly suppress (Can they actually suppress normally in conventional in vitro suppression assays?) and why specifically not type 1 immune cells. More data addressing this question would certainly increase the scope of the paper immensely.

In addition to the dysregulation of type 1 IFN γ -producing effector T cells in *FYC1/2* mice, we also observe greatly elevated levels of serum IgE, and increased proportions and numbers of both germinal centre B cells and T follicular helper cells in *FYC1/2* mice, which shows that type 2 immunity is also affected. Furthermore, we also observe a selective increase in the number of "resident" cDC2 in the LN of *FYC1/2* mice. As cDC2 can prime Th17 and Th2 responses (<https://doi.org/10.1038/s41423-021-00741-5>), this suggests the dysregulation is not limited to type 1 responses.

To test if Tregs lacking *Zfp3611* and *Zfp3612* can suppress normally in a conventional *in vitro* suppression assay we performed this assay and observed no difference between control Tregs and Tregs lacking *Zfp3611* and *Zfp3612* in their ability to suppress the proliferation of CD4⁺ conventional T cells or CD8⁺ T cells *in vitro* (Reviewer Figure 7).

Reviewer Figure 7. Tregs lacking *Zfp3611* and *Zfp3612* suppress the proliferation of CD4 T cells *in vitro*. Tregs from *Cd4cre I112* mice were sorted (CD4⁺CD25⁺). 50000 Naive CD4⁺ T cells were labelled with Cell Trace Violet and activated with anti-CD3 (1ug/ml) and anti-CD28 (0.5ug/ml) antibodies. CD4⁺ T cells were co-cultured with 50,000 Splenocytes from *Rag2*-deficient mice and different numbers of Tregs (50000;25000;12500). FACS analysis was performed at day three of culture.

Minor comment on Figure 7i and Sup. Fig. 8. The staining of RORγt⁺ Tregs doesn't really show any pos. population to me. I suggest to stain for these Treg subsets (also the GATA3⁺ Treg) in intestinal Treg cells rather than Treg cells from Spleen/LN.

We have now revised the visualisation of the flow cytometry data in **Figure 7** and **Supplementary Fig.S18e** to display larger dots for each event and added the number of events for each data file. In addition, we include new data showing RORγt expression in CD4⁺ cells from mesenteric LN (**Supplementary Fig. S18e**). Together, we think this revised figure more clearly presents the findings.

Reviewer #3 (Remarks to the Author)

In this study, Saenz-Narciso and colleagues investigate the role of ZFP36 family RBPs in Tregs. They specifically ablate *Zfp3611*, *Zfp3612*, and *Zfp3611/2* in Tregs and find that the loss of either protein or the combined loss of both leads to autoimmunity, as evidenced by the expansion of activated CD4 and CD8 T cells expressing proinflammatory cytokines. Mechanistically, these RBPs regulated multiple aspects of Treg biology, including endocytosis and sensitivity to cytokines, both of which had significant implications for Treg stability and function. The authors further demonstrated that IFNγ drives the inflammatory phenotype in mice lacking *Zfp3611/2* specifically in Tregs, noting that deletion of one allele of IFNγ is sufficient to ameliorate this phenotype. This is an excellent study with high-quality data; however, a few issues require clarification.

We thank the reviewer for their positive comments and feedback on our manuscript which we have taken on board to improve the study and its presentation, we respond to each point below.

Revisions/new text are highlighted in the revised manuscript.

1. The authors state that only 9% of *FYCI1* and 13% of *FYCI1I2* mice develop autoimmunity. Why is the disease penetrance not 100%?

We would like to clarify that all the male mice analysed, even those that did not display overt clinical signs, showed increased lymph node cellularity, increased proportions and numbers of effector cells and an increased proportion of IFN γ -producing Treg. The penetrance of these phenotypes is 100%. We now include data showing lymph node cellularity is increased in mice with clinical signs (Supplementary Fig. 1d). The reason why only some mice go on to display clinical signs that exceed the severity limit is uncertain. It is likely that the variable tendency of male mice to fight when cohoused induces local infection and immune responses which may trigger clinical disease in some instances.

It appears the authors focus solely on male mice. Is there a specific reason for not analysing homozygous knockout females?

In order to generate homozygous female mice, we would need to cross hemizygous males (*FYC1I2* which have the potential to develop clinical symptoms) with heterozygous females. We tried breeding from hemizygous males crossed to heterozygous females but were unsuccessful in generating homozygous females (from two pairs and 30 pups which survived to weaning, only 1 homozygous female was identified from 10 females) so we did not continue for practical and welfare reasons.

2. In female *FYCI1I2/+* mice, the *Foxp3+YFP+* population was outcompeted by *Foxp3+YFP-WT* Tregs, suggesting that Tregs lacking *I1I2* have reduced competitive fitness. Do knockout Tregs exhibit an activated phenotype in this context (with competition), or is there a skewed nTreg/eTreg ratio?

We have now included data for the percentage of nTreg and eTreg in female mice; the ratio between FOXP3⁺ cre-negative and FOXP3⁺ cre-positive Treg is not different between *FYC⁺ I1I2* and *FYC⁺* mice (Supplementary Fig. 3a).

3. At the RNA level, there is downregulation of *Ctla4*, which appears to correlate with low protein levels of total CTLA4, despite the claim of endocytosis. What is the underlying cause of the downregulation of CTLA4 protein?

In Fig. 2b of the original manuscript the labelling of the MA plot showing the position of *Ctla4* mRNA may have suggested that its amounts are decreased, but this is not the case. Our data show that *Ctla4* is not decreased at the RNA level in *FYC⁺ I1I2* mice. For clarity, we now show in Supplementary Fig. 8a of the manuscript no difference in the normalised read counts or splicing of the *Ctla4* transcript in Treg from *FYC⁺ I1I2* or *FYC⁺* mice.

In Fig. 3e of the original and revised manuscript we report a minor decrease (1.5-fold) in total CTLA-4 protein (geoMFI) detected by intracellular flow cytometry in the CD62L^{hi} nTreg subset in *FYC⁺ I1I2* mice. We wish to emphasise that these cells were cultured *in vitro* in the presence of 2ng/ml IL-2.

In the original manuscript Supplementary Fig. 4a showed intracellular CTLA-4 staining in samples that were fixed directly *ex vivo*. This showed no difference in the total CTLA-4 protein in YFP⁺ nTreg from *FYC⁺ I1I2* mice compared to control *FYC⁺* mice – the subset in which we observed the decreased cycling of CTLA-4. In the revised manuscript this data is now shown in Supplementary Fig. 6a. In the eTreg population (where there are no defects in cycling) we observed a very minor decrease (1.3-fold) in CTLA-4 geoMFI (Supplementary Fig. 6a). Many factors could be affecting the expression of CTLA-4, e.g. TCR signalling, FOXP3 and IL-2 upregulate CTLA-4 expression.

Following internalisation, CTLA-4 can also be delivered to lysosomes where it is degraded.

4. Is defective STAT5 signalling responsible for the lack of competitive fitness in FYC112 mice?

We suggest that the most clearcut way to test this hypothesis would be to introduce a constitutively active STAT5 transgene which is predicted to rescue the phenotype. Because the STAT5 transgene causes leukaemia this would have to be achieved using the conditional allele described here (<https://pmc.ncbi.nlm.nih.gov/articles/PMC5071159/>). We do not have these mice and, time and resource consideration make this experiment unfeasible.

Can the IL-2/Ab complex expand YFP+ Tregs in FYC112/+ mice?

Based on the response of the cKO Treg from *FYC⁺ 112* mice to IL-2 *in vitro*, and the expansion of cKO Treg in male mice, it is reasonable to expect the cKO Treg in *FYC⁺ 112* mice will respond if enough stimulus is provided. After considering this carefully, we have concluded that the substantial numbers of animals required to optimise dosing with the IL-2/Ab complex and subsequently to demonstrate a difference in the response to IL-2 *in vivo* is hard to justify against the limited mechanistic insight it will provide.

5. FYC112 Tregs produce more IFN γ and IL-10 when stimulated. Is this phenotype a result of inflammation in these mice? Would YFP+ Tregs in FYC112/+ mice display the same phenotype? A major concern is distinguishing the actual phenotype of Tregs caused by 112 deletion from that induced by inflammation. For example, bulk RNA sequencing has been performed on Tregs from FYC112/+ mice (with no inflammation), while scRNA sequencing was conducted on Tregs from FYC112 mice (where inflammation is high). Can these two datasets be compared (using scRNA-seq pseudobulk analysis) to differentiate between inflammation-induced and genuine 112-dependent gene expression?

The analysis of female mice heterozygous for the *FYC* allele is used to identify cell intrinsic effects in Tregs caused by lack of the RBP in the absence of inflammation. We have measured IFN γ and IL-10 produced by Treg in these mice.

Analysis of IFN γ produced by Tregs in *FYC⁺ 112* mice revealed a higher proportion of YFP+ Treg that were IFN γ + compared to either YFP+ Treg in *FYC⁺* mice (three-fold increase) or cre-negative cells in the same mouse (**Supplementary Fig. 19a**). In addition, we observed higher amounts (3.6-fold increase) of *Ifng* at the RNA level in the RNAseq data from YFP+ nTreg from *FYC⁺ 112* mice compared to YFP+ nTreg from *FYC⁺* mice (**Supplementary Fig. 19b**).

This reflects a cell intrinsic effect meaning that the increase in proportion of IFN γ + Treg observed in *FYC* male mice is not solely caused by inflammation. Nonetheless, the increase in proportion is smaller in the absence of inflammation compared with the increase observed in male mice (five-fold increase; **Figure 7I**), indicating that the phenotype is exacerbated by the inflammatory environment.

We observed no difference in proportions of IL-10+ cells between YFP+ Treg from female *FYC⁺* and *FYC⁺ 112* mice (**Supplementary Fig. S19c**). These data suggest that the increased proportions (two-fold increase) of IL-10+ *FYC 112* Tregs are likely a result of the inflammation in male mice.

As the reviewer states, the bulk RNA-seq is performed in the absence of inflammation, and thus the findings from this data show the cell-intrinsic effects of the deletion independently of inflammation. Pseudobulk analysis of the scRNA-seq data, which was performed on cells in inflammatory conditions, would, at best, serve only to indicate additional pathways that are downstream of inflammation. However, differences in the cell populations sorted for the two experiments, and in the sensitivity of pseudobulk compared with bulk RNA-seq analysis of cells from females will together add substantial noise to the data, making it very difficult to draw robust and meaningful conclusions from the analysis. This would not clarify the cell-intrinsic effects of *Zfp361/112* deletion.

One of the major findings of the scRNA-seq was an increased proportion of cells displaying an IFN γ signature. Our analysis of the bulk RNA-seq data indicates that this phenotype is not driven solely by the inflammatory environment, since gene set enrichment analysis showed a trend (albeit not quite reaching significance) towards increased expression of genes that are increased upon IFN γ treatment (**Fig. 2g**). This is also supported by our data showing increased responsiveness to IFN γ in the absence of inflammation (**Fig. 6**). In addition to the increased expression of *Ifng* at the RNA and protein level in female heterozygous mice (see above), we also saw non-significant trends towards increased mRNA abundance of *Gata3* and *Cxcr3*, and decreased *Rorc* mRNA (**Reviewer Figure 8**). This adds further evidence that the increased frequency of cells producing IFN γ is, in part, a cell intrinsic effect.

Reviewer Figure 8. Normalised read counts for *Ifng*, *Cxcr3*, *Gata3* and *Rorc* comparing nTreg from *FYC*⁺ and *FYC*⁺ *I1I2* mice. Bars indicate the mean, and points indicate normalised counts for individual female mice analysed by bulk RNA-seq. Numbers above indicate FDR-adjusted p values; these and normalised counts were derived from DESeq2 analysis

Our scRNA-seq data also allowed us to better define the skewing of the Tregs towards a more activated phenotype, characterising the transcriptome of the populations that are gained and lost by the *FYC I1I2* mice. In contrast to male *FYC I1I2* mice (**Fig. 1c**), the female heterozygous mice do not show an increase in the proportion of *icre*⁺ Tregs with an effector phenotype: we now show this data in **Supplementary Fig. 3a**. Thus, the expansion of effector relative to naive Tregs is driven by the inflammatory environment, and we would not expect to see the same level of skewing in the transcriptome of Tregs from female heterozygous mice. Our bulk RNA-seq was also performed only on naive Tregs, due to the difficulty in obtaining sufficient numbers of effector Tregs, limiting our capacity to draw conclusions about the effector population. Nonetheless, we are able to see some trends in our bulk RNA-seq that align with our findings in the scRNA-seq. Gene set enrichment analysis using the marker genes for each of the clusters identified in the scRNA-seq shows that, in the bulk RNA-seq, *FYC*⁺ *I1I2* Tregs are strongly enriched for genes that are highly expressed in cluster 5 (**Reviewer Figure 9**). This cluster represents naive Tregs (**Supplementary Fig. 18b**) but, in contrast to clusters 0 and 2, it is maintained in *FYC I1I2* mice. Whilst cluster 0 markers are also enriched in knockout Tregs in the bulk RNA-seq, these comprise only 5 genes, all of which are also markers for cluster 5. This suggests that the naive Tregs in female heterozygous mice already show some skewing towards this phenotype, although we are unable to draw conclusions as to the magnitude of this skewing. Of note, the cluster 5 markers include numerous genes that are known to be induced by IFN γ , such as *Ly6a*, *Ly6c1*, *ligp1*, *Dapl1*, *Igtp* and *Samhd1*, and over half of the *FYC I1I2* cells within this cluster were defined as having an

IFN γ gene signature (**Fig. 7e**), suggesting that the skewing towards this phenotype may result, at least in part, from the increased responsiveness to IFN γ signalling.

Reviewer Figure 9. GSEA analysis in bulk RNAseq from nTreg from *FYC*⁺ *I112* females showed an enrichment in expression of cluster markers which defined cluster 5 in scRNAseq from *FYC* *I112* male mice. GSEA was performed to assess enrichment in *FYC*⁺ *I112* compared with *FYC*⁺ nTreg, using the marker genes defining each of the clusters identified in scRNAseq analysis of *FYC* *I112* Treg. X axis represents $-\log_{10}(\text{FDR-adjusted p value})$, and colour represents normalised enrichment score from the GSEA analysis. Numbers to the right of each bar indicate the total number of marker genes for each cluster.

In addition to the enrichment of cluster 5 marker genes, we also saw a significant enrichment of cluster 4 markers within the *FYC*⁺ *I112* naive Tregs. This represents effector Tregs in the scRNA-seq and is the cluster containing the highest proportion of *FYC* *I112* (relative to *FYC*) Tregs. This suggests that the naive Tregs in the heterozygous females already show some increased expression of genes characteristic of effector Tregs; however, our flow cytometry data show that this is insufficient to cause expansion of a *bona fide* effector Treg population.

6. What happens to Tregs in non-lymphoid tissues? Are there any defects in migration?

We have data from the liver and lamina propria from male mice showing the numbers of CD4⁺YFP⁺ Nrp1⁺ Treg are not different between *FYC* and *FYC* *I112* mice (**Reviewer Figure 10**). We conclude that Treg are present in these non-lymphoid tissues. We have not measured migration directly but our data show that Tregs are present in spleen, lymph nodes, thymus and non-lymphoid tissue.

Reviewer Figure 10. *FYC* *I112* Tregs are present in liver and lamina propria. Representative flow cytometry plots showing the expression of NRP1 versus CD62L on TCR β ⁺CD4⁺ YFP⁺ Treg in the lamina propria (upper panels) and liver (bottom panels) from *FYC* and *FYC* *I112* mice; including proportions and numbers of CD62L⁺ NRP1⁺ Tregs in each tissue.

Final response to reviewer's comments

Reviewer #1 (Remarks to the Author):

The authors have adequately addressed all of my points and provided important new information.

We thank the reviewer for their positive comments. No additional material was requested.

Reviewer #2 (Remarks to the Author):

The authors have addressed all my concerns. Thank you.

Regarding Figure 1, I would suggest to include the histology data (now Sup. Fig. 1c) and the LN cellularity data (now Sup. Fig. 1d and 2a) into the main Figure, bc they are strong phenotypic data that don't need to be hidden in the supplement.

We thank the reviewer for their positive comments.

These data are now included in the main Figure 1b,c.

The potential reasons why only a minor fraction of mice develops clinically symptoms, should be briefly discussed in the manuscript.

This is briefly discussed in the Discussion under "Limitations of the study".

Reviewer #3 (Remarks to the Author):

The authors have addressed all of my queries. I have no further concerns. This is an excellent study!

We thank the reviewer for their positive comments. No additional material was requested.